# Ordering-Based Causal Discovery for Linear and Nonlinear Relations

**Zhuopeng Xu    Yujie Li    Cheng Liu    Ning Gui**[*]
School of Computer Science and Engineering
Central South University
{xuzhuopeng, yujieli}@csu.edu.cn, {liuchengstudy, ninggui}@gmail.com

## Abstract

Identifying causal relations from purely observational data typically requires additional assumptions on relations and/or noise. Most current methods restrict their analysis to datasets that are assumed to have pure linear or nonlinear relations, which is often not reflective of real-world datasets that contain a combination of both. This paper presents CaPS, an ordering-based causal discovery algorithm that effectively handles linear and nonlinear relations. CaPS introduces a novel identification criterion for topological ordering and incorporates the concept of "parent score" during the post-processing optimization stage. These scores quantify the strength of the average causal effect, helping to accelerate the pruning process and correct inaccurate predictions in the pruning step. Experimental results demonstrate that our proposed solutions outperform state-of-the-art baselines on synthetic data with varying ratios of linear and nonlinear relations. The results obtained from real-world data also support the competitiveness of CaPS. Code and datasets are available at `https://github.com/E2real/CaPS`.

## 1 Introduction

Causal discovery uncovers latent causal relationships within data by modeling a Directed Acyclic Graph (DAG) connecting various variables. This field is of significant importance in domains such as biology [1], epidemiology [2], and finance [3]. Due to the considerable expense associated with conducting interventional experiments, the recent emphasis on causal discovery has gradually shifted from discovery with interventional data [4, 5, 6] to discovery solely based on observational data.

In general, the problem of causal discovery from observational data faces the identifiability issue. Different generative models with different causal relations might produce the same data distribution. Many recent works try to have uniquely identified DAG by placing different types of assumptions on the noise and/or relations, e.g., Shimizu et al. [7] prove that linear causal relations with non-Gaussian additive noise can be identifiable; Peters and Bühlmann [8] prove that Gaussian linear structural equation models (SEMs) with equal variances are identifiable. For nonlinear causal relations, Peters et al. [9] relax the assumption of noise and proves the identifiability of DAG. From the discussion mentioned above, existing approaches normally limit their discussions to distributions with either pure linear or pure nonlinear relations.

However, real-world data often contain both types of causal relations and run against their basic assumptions. These approaches work well when the observational data match their prespecified (non-)linear or nonlinear relations while suffering significant performance loss when their assumptions mismatch. Fig. 1 illustrates the performance of three solutions: GOLEM [10] for linear relations, SCORE [11] for nonlinear relations, and our proposed CaPS, on synthetic data with varying proportions of linear relations. The performance of SCORE decreases as the linear ratio increases, while

---

[*]Corresponding author.

38th Conference on Neural Information Processing Systems (NeurIPS 2024).

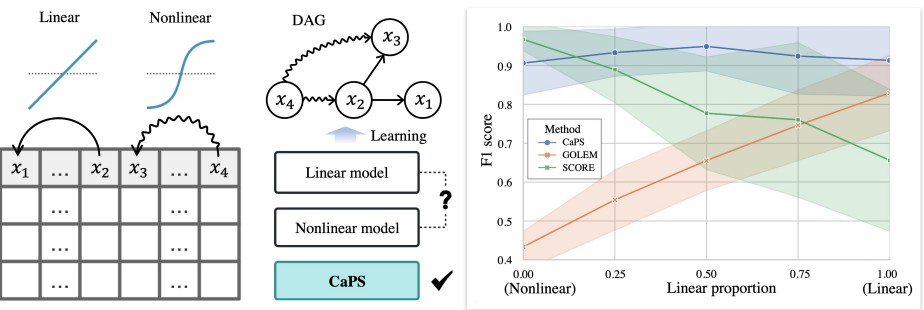

Figure 1: Performance of different solutions under datasets with different linear proportions. Since we don't know whether the real data is linear or nonlinear, it is difficult to choose an effective model. Thus, we need a method that works well in both linear and nonlinear and most possibly mixed cases.

GOLEM performs poorly when the linear relation ratio is low. This indicates that approaches with strong restrictions on the types of relations are not suitable for real-world applications. Consequently, it is necessary to find a unified causal learning framework to capture both types of relations.

Ordering-based methods [12] divide the casual discovery process into two subtasks: (i) topological ordering and (ii) post-processing. It has been shown to have the capability to reduce the complexity of DAG discovery while keeping the acyclicity constraint. In addition, ordering-based methods guide the direction of causal relations, thus avoiding the fitting of a potential inverse model. However, existing ordering-based approaches still face the same problem of relying on the assumption of linear, e.g., LISTEN[13] or nonlinear causal relations, e.g., SCORE.

This paper introduces a unified approach, Causal Discovery with Parent Score (CaPS), that does not rely on linear or nonlinear assumptions, within the scope of ordering-based casual discovery . To determine the topological ordering, we propose a novel unified criterion for distinguishing leaf nodes by utilizing the expectation of the Hessian of the data log-likelihood. Additionally, inspired by the average treatment effect (ATE) in estimating causal effects [14], the parent score, a new metric, is proposed to represent the average causal effects of all samples. No matter whether the causal relations are linear or nonlinear, this metric can be used to effectively guide parent selection in post-processing.

**Contributions.** 1) A novel ordering criterion is proposed for distinguishing leaf nodes, which enables learning of topological ordering for data with both types of relations; 2) A new criterion, parent score is introduced to reflects the strength of the average causal effect of a given parent. Utilizing this criterion, CaPS designs pre-pruning and edge supplement operations to speed up the pruning process and rectify inaccurate predictions in the pruning step. 3) Extensive experiments are conducted to compare eight state-of-the-art baselines on synthetic and real-world data, showing competitive results. Furthermore, various analysis simulations demonstrate the effectiveness of our proposed designs.

## 2 Related Work

**Causal discovery for SEMs.** Gaining knowledge of DAGs from observational data often necessitates additional suppositions about the distributions and/or relations. This paper discusses SEM-based studies from either the linear or nonlinear point of view.

To ensure the identifiability of linear causal relations, additional noise constraints must be made, for example, Gaussian noise with equal variance [8] or conditional variance [13]. Earlier work by LiNGAM [7] demonstrated the identifiability of linear causal relations with non-Gaussian noise and proposed an ICA-based method to identify this SEM. NOTEARS [15] proposed a novel continuous optimization approach based on the trace exponential function, which provides a new idea of continuous optimization for causal discovery. GOLEM further [10] proposed a continuous likelihood-based method with soft sparsity and DAG constraints, which improved the performance of linear causal discovery. The optimization of these works is largely dependent on the linear parameterization of the weighted adjacency matrix.

For nonlinear SEMs, e.g., the nonlinear additive noise model (ANM) [9] is known to be identifiable with arbitrary additive noise, except in some rare cases. To discover causal relations under nonlinear SEM, many previous works [16, 17, 18, 19, 20] proposed different DAG learning methods with a continuous acyclicity constraint. These works generally require a training model with an augmented Lagrangian approach which results in considerable computation cost.

**Ordering-based causal discovery** can partially avoid the aforementioned problems, as the order space is much smaller than the DAG space. CAM [21] is an early ordering-based approach that uses a greedy search to estimate the topological ordering and significance tests to prune the DAG. LISTEN [13] proposed a new method for distinguishing leaf nodes of linear causal relations, based on the precision matrix [22]. However, these methods are based on the connection between the precision matrix and the weight matrix, which is not a unified criterion for linear and nonlinear SEMs. SCORE [11] estimated the score (the Jacobian of logarithmic probability of data, $\nabla \log p(x)$) using the second-order Stein gradient estimator, and then determined the leaf nodes based on the score to find the topological ordering. DiffAN [23] was proposed to estimate the score via a diffusion model.

Several recent works merge the two-stage process into one step by end-to-end differentiable optimizations to address the issue of error propagation, e.g VI-DP-DAG [24] with variational inference and DAGuerreotype [25] over the polytope. DAGuerreotype provides two variants specifically designed for linear/nonlinear relations. However, none of the above works provides a uniform solution that can identify DAG in datasets with both linear and nonlinear relations.

## 3 Preliminaries

### 3.1 Structural Equation Model

The structural equation model for the causal discovery can be represented as : for random variable $\mathbf{x} = \{x_1, x_2, ..., x_d\} \in \mathbb{R}^d$ sampling from the real joint probability distribution $p(\mathbf{x})$, we want to find a faithful causal graph $\mathcal{G}$ to represent the causal relationships between variables of different dimensions. The SEM is defined with equation (1):

$$x_i = f_i(pa_i(x)) + \epsilon_i \tag{1}$$

where $x_i \in \mathbf{x}$, $i = 1, 2, ..., d$. $pa_i(x)$ denotes the parents of $x_i$, $f_i$ denotes the causal function, and $\epsilon_i$ denotes the additive noise of $x_i$. For each $x_i$, the parents and the noise are independent of each other, $pa_i(x) \perp\!\!\!\perp \epsilon_i$, and there is no unobservable confounder. Each causal function $f_i$ can be linear and nonlinear, and the additive noise $\epsilon_i \sim \mathcal{N}(0, \sigma_i^2)$ is Gaussian. These are the basic assumptions of ANM [9], and we relax the linear and nonlinear conditions in our SEM.

### 3.2 Topological Ordering

Finding topological ordering is an important subtask for ordering-based causal discovery, which can reduce the DAG search space.

**Definition.** Since the topological ordering of the causal graph $\mathcal{G}$ may not be unique, we define a set of order permutations $\Pi$ to represent all valid topological orderings. For any order permutation $\pi \in \Pi$, a parent node must always be before a child node, i.e. $\pi(i) < \pi(j)$ if $x_j$ is a descendant of $x_i$ on $\mathcal{G}$. The corresponding initialized adjacency matrix $\mathcal{A}$ should have $\mathcal{A}_{i,j} = 1$ and $\mathcal{A}_{j,i} = 0$.

**Estimation.** To identify the leaf nodes of a DAG, SCORE suggests using the score $s_j(x)$, which is equal to the logarithmic gradient of the joint probability distribution $p(\mathbf{x})$ with respect to $x_j$. If a Markov chain is used to represent $p(\mathbf{x})$, then the score of each variable is equivalent to the logarithm gradient of the joint probability distribution.

$$
\begin{aligned}
s_j(x) &= \nabla_{x_j} \prod_{i=1}^{d} p(x_i | pa_i(x)) = \nabla_{x_j} \sum_{i=1}^{d} \log p(x_i | pa_i(x)) \\
&\overset{(i)}{=} \nabla_{x_j} [-\frac{1}{2} \sum_{i=1}^{d} (\frac{x_i - f_i(pa_i(x))}{\sigma_i})^2 - \frac{1}{2} \sum_{i=1}^{d} \log(2\pi\sigma_i^2)] \\
&= -\frac{x_j - f_j(pa_j(x))}{\sigma_j^2} + \sum_{i \in ch(j)} \frac{\partial f_i}{\partial x_j}(pa_i(x)) \frac{x_i - f_i(pa_i(x))}{\sigma_i^2}
\end{aligned}
\tag{2}
$$

where $(i)$ uses the *change of variables theorem* with $x_i - f_i(pa_i(x)) = \epsilon_i$ and $ch(j)$ denotes the children of $x_j$ in the causal graph $\mathcal{G}$. According to Eq.2, for each leaf node, it is easy to see that $\frac{\partial s_j(x)}{\partial x_j} = -\frac{1}{\sigma_j^2}$ is a constant. Thus, given the nonlinear assumption of the causal function $f_i$, SCORE has shown that $x_j$ is a leaf node iff the variance of $\frac{\partial s_j(x)}{\partial x_j}$ is zero. The score $s_j(x)$ and each element on the diagonal of the score's Jacobian $\frac{\partial s_j(x)}{\partial x_j}$ can be estimated using the second-order Stein gradient estimator [26] or the diffusion model [23].

## 4 Causal Discovery with Parent Score

This section first examines the limits of current baselines under non-preassumed relations. Then, the design of CaPS in both topological ordering and post-processing is introduced.

### 4.1 Leaf Nodes Discrimination

Previous attempts to order nodes according to a certain criterion were unsuccessful due to the lack of a unified standard to distinguish leaf nodes in datasets with mix relations. To make this point more evident, we provide examples of both linear and nonlinear cases. LISTEN [13] for linear causal relations uses the minimum value of the precision matrix's diagonal to distinguish leaf nodes. However, this approach fails because the connection between the precision matrix and the true causal graph no longer holds under nonlinear causal relations, which is detailed in Appendix A.1.

SCORE for nonlinear causal relations cannot differentiate leaf nodes in linear causal relations. To illustrate this, consider a simple linear causal case $x_i = \sum_{x_k \in pa_i(x)} w_{i,k} x_k + \epsilon_i$, where $f_i$ is linear in ANM. In this linear SEM, $\frac{\partial f_i}{\partial x_j}(pa_i(x)) = w_{i,j}$ is a constant and $\frac{\partial^2 f_i}{\partial x_j^2}(pa_i(x)) = 0$. Consequently, for any node, the value of each element on the diagonal of the score's Jacobian is always constant, i.e., $\frac{\partial s_j(x)}{\partial x_j} = -\frac{1}{\sigma_j^2} - \sum_{i \in ch(j)} \frac{w_{i,j}^2}{\sigma_i^2}$, making SCORE unable to differentiate leaf nodes. To address this issue, we propose a new discriminant criterion in Theorem 1 effective in both linear and nonlinear causal relations and give its sufficient conditions for identifiability in Assumption 1.

**Assumption 1.** *(Sufficient conditions for identifiability). The topological ordering of a causal graph is identifiable if **one** of the following sufficient conditions is satisfied.*

*(i) **Non-decreasing variance of noises.** For any two noises $\epsilon_i$ and $\epsilon_j$, $\sigma_j \geq \sigma_i$ if $\pi(i) < \pi(j)$.*

*(ii) **Non-weak causal effect.** For any non-leaf nodes $x_j$, $\sum_{i \in Ch(j)} \frac{1}{\sigma_i^2} \mathbb{E}[(\frac{\partial f_i}{\partial x_j}(pa_i(x)))^2] \geq \frac{1}{\sigma_{min}^2} - \frac{1}{\sigma_j^2}$.*

where $\sigma_{\min}$ is the minimum variance for all noises. Assumption 1 gives two conditions for identifiability which no longer depends on the linearity or nonlinearity of the causal function and relaxes previous identifiability conditions. Condition $(i)$ is an extension of the equal variance assumption [8, 13]. Condition $(ii)$ is a new sufficient condition, which first quantitatively gives a lower bound of identifiable causal effects. It enables CaPS to identify causal relations even in scenarios outside of non-decreasing variance. For example, considering a variance-unsortable scenario with $\sigma^2 \sim U(0.1, 1)$ and causal effect greater than 3, CaPS can also work well because the the sum of parent score is greater than the given lower bound in condition$(ii)$. The meaning of condition $(ii)$ will be further discussed in section 4.2. Under conditions $(i)$ or $(ii)$, the topological ordering can be identified by Theorem 1.

**Theorem 1.** *Let $s(x) = \nabla \log p(x)$ be the score and let $diag(\cdot)$ be the diagonal elements of the matrix. For any $x_j$ in the causal graph $\mathcal{G}$:*

$$j = \arg\max(diag(\mathbb{E}[\tfrac{\partial s(x)}{\partial x}])) \Rightarrow x_j \text{ is a leaf node}$$

*Proof.* (Simplified version; details are in Appendix A.2.)

For an arbitrary node $x_j$ in the causal graph $\mathcal{G}$, we focus on the diagonal of the score's Jacobian.

$$\frac{\partial s_j(x)}{\partial x_j} = -\frac{1}{\sigma_j^2} - \sum_{i \in ch(j)} \frac{1}{\sigma_i^2}(\frac{\partial f_i}{\partial x_j}(pa_i(x)))^2 + \sum_{i \in ch(j)} \frac{\partial^2 f_i}{\partial x_j^2}(pa_i(x)) \cdot \frac{x_i - f_i(pa_i(x))}{\sigma_i^2} \qquad (3)$$

Since $\frac{x_i - f_i(pa_i(x))}{\sigma_i^2} = \frac{\epsilon_i}{\sigma_i^2} \sim \mathcal{N}(0, \frac{1}{\sigma_i^2})$ and $pa_i(x) \perp\!\!\!\perp \epsilon_i$ in our SEM, the expectation of $\frac{\partial s_j(x)}{\partial x_j}$ can be restated as:

$$\mathbb{E}[\frac{\partial s_j(x)}{\partial x_j}] = -\frac{1}{\sigma_j^2} - \sum_{i \in Ch(j)} \frac{1}{\sigma_i^2} \mathbb{E}[(\frac{\partial f_i}{\partial x_j}(pa_i(x)))^2] \tag{4}$$

Suppose that $x_l$ is a leaf node and $x_n$ is a non-leaf node, we have $\mathbb{E}[\frac{\partial s_l(x)}{\partial x_l}] = -\frac{1}{\sigma_l^2}$. Then, $\mathbb{E}[\frac{\partial s_l(x)}{\partial x_l}] \geq \mathbb{E}[\frac{\partial s_n(x)}{\partial x_n}]$ always holds under conditions $(i)$ or $(ii)$. Thus, the node in $\arg\max(\mathrm{diag}(\mathbb{E}[\frac{\partial s(x)}{\partial x}]))$ will always be the leaf node. □

Theorem 1 suggests that the expectation of the diagonal of the score's Jacobian can be used to identify leaf nodes. Thus, applying Theorem 1, the topological ordering is identifiable by iteratively eliminating the current leaf node [27]. The detailed procedure is included in Algorithm 1, where we use the second-order Stein gradient estimator to estimate the score's Jacobian $\frac{\partial s(x)}{\partial x}$.

## 4.2 Parent Score

Theorem 1 specifies how to identify the correct topological ordering of the graph $\mathcal{G}$. However, it is not straightforward to determine the parents of each node. To identify the true causal graph $\mathcal{G}$, we need information beyond permutation to guide the selection of parents. Thus, we need a metric that can qualitatively express the causual effects from a parent node to one of its children. Here, a new metric, "Parent Score" is proposed to approximate the average causal effect.

**Definition of parent score.** We define Eq.5 to express the parent score $\mathcal{P}_{i,j}$ which approximates the strength of the average causal effect of all samples from $x_j$ to $x_i$.

$$\mathcal{P}_{i,j} = \begin{cases} \frac{1}{\sigma_i^2} \mathbb{E}[(\frac{\partial f_i}{\partial x_j}(pa_i(x)))^2], & x_j \in pa_i(x) \\ 0, x_j \notin pa_i(x) \end{cases} \tag{5}$$

where $\mathcal{P}_{i,j} = 0$ if $x_j$ is not a parent of $x_i$, and $\mathcal{P}_{i,j} > 0$ if $x_j$ is a parent of $x_i$.

To illustrate the meaning of this definition, we propose a new metric of the Squared Average Treatment Effect (SATE) extended from Average Treatment Effect (ATE, $\mathbb{E}[Y^{(T=1)} - Y^{(T=0)}]$). In the task of estimating causal effects [14], ATE is often used to measure the average effect of a treatment or intervention on an outcome variable. Since we focus only on the strength of the effect rather than on the positive or negative effect, SATE is defined as follows:

$$\mathrm{SATE}_i^j = \mathbb{E}[(x_i^{(T_j=1)} - x_i^{(T_j=0)})^2] \tag{6}$$

where $T_j = 1$ and $T_j = 0$ indicate whether $x_j$ are treated or not. With a small additive treatment, we show that SATE from $x_j$ to $x_i$ can be approximated by $\mathbb{E}[(\frac{\partial f_i}{\partial x_j}(pa_i(x)))^2]$. Thus, the parent score $\mathcal{P}_{i,j}$ is the causal effect of the parent scaled by its variance of noise. The detailed derivation is shown in Appendix A.3.

**Computing parent score.** Parent score cannot be obtained directly from the summation of average causal effects on childrens in Eq.4, thus we propose an iterative decoupling process and define

$$\mathcal{J} = \{\mathbb{E}[\frac{\partial s_1(x)}{\partial x_1}], \mathbb{E}[\frac{\partial s_2(x)}{\partial x_2}], ..., \mathbb{E}[\frac{\partial s_d(x)}{\partial x_d}]\} \tag{7}$$

to denote the expectation of the diagonal of the score's Jacobian. By removing the node $x_i$, a new vector $\mathcal{J}_{-i}$ is defined as follows:

$$\mathcal{J}_{-i} = \{\mathbb{E}[\frac{\partial s_1(x_{-i})}{\partial x_1}], \mathbb{E}[\frac{\partial s_2(x_{-i})}{\partial x_2}], ..., \mathbb{E}[\frac{\partial s_i(x)}{\partial x_i}], ..., \mathbb{E}[\frac{\partial s_d(x_{-i})}{\partial x_d}]\} \tag{8}$$

where $x_{-i}$ represents the remaining data after removing the feature of $i$-th dimension. For the $i$-th element of $\mathcal{J}_{-i}$, we fill the $i$-th element of $\mathcal{J}$. Then, each row vector of the matrix of parent score $\mathcal{P} \in \mathbb{R}^{d \times d}$ is equivalent to:

$$\mathcal{P}_{i,:} = \mathcal{J}_{-i} - \mathcal{J} \tag{9}$$

The complete $\mathcal{P}$ can be obtained by iteratively computing parent score of each row. The specific derivation of this procedure is given in Appendix A.4. The Algorithm 1 describes the process of

---

**Algorithm 1** Ordering and Computing parent score

---

**Input**: data matirx $X \in \mathbb{R}^{n \times d}$
**Output**: permutation $\pi$, parent score $\mathcal{P}$
 1: initialize $\pi \leftarrow [\,]$, removed set $r \leftarrow [\,]$, nodes $\leftarrow [1, 2, ..., d]$, $\mathcal{P} \leftarrow \mathbf{0}$
 2: estimate $\frac{\partial s(X)}{\partial x}$ and then obtain $\mathcal{J}$ using Eq.7
 3: **for** $i = 1, 2, ..., d$ **do**
 4:     estimate $\frac{\partial s(X_{-i})}{\partial x}$ and $\frac{\partial s(X_{-r})}{\partial x}$
 5:     obtain $\mathcal{J}_{-i}$ using Eq.8 and then compute parent score $\mathcal{P}_{i,:} \leftarrow \mathcal{J}_{-i} - \mathcal{J}$ using Eq.9
 6:     distinguish current leaf node $l \leftarrow$ nodes$[\arg \max \operatorname{diag}(\mathbb{E}[\frac{\partial s(X_{-r})}{\partial x}])]$ using Theorem.1
 7:     update $\pi \leftarrow [l, \pi]$, $r \leftarrow r + \{l\}$, nodes $\leftarrow$ nodes $- \{l\}$
 8: **end for**
 9: **return** $\pi, \mathcal{P}$

---

finding the topological order and computing the parent score, where $X_{-i}$ denotes the data matrix with the $i$-th feature removed. Similarly, $X_{-r}$ denotes the data matrix with a set of removed features.

**Association with leaf nodes discrimination.** We can revisit the criterion to distinguish leaf nodes with parent score. According to the definition of parent score, given the node $x_j$, $\sum_{i=0}^{d} \mathcal{P}_{i,j}$ can be considered as the total causal effect to its children. The sufficient condition $(ii)$ for identifiability can be restated as $\sum_{i=0}^{d} \mathcal{P}_{i,j} \geq \frac{1}{\sigma_{\min}} - \frac{1}{\sigma_j}$ when $x_j$ is not a leaf. It means that the causal relations can be identified if the causal effect stronger than the given lower bound, which gives a quantitative interpretation for an intuitive conclusion. Under this sufficient condition, the meaning of Theorem 1 can be further explained in the following corollary.

**Corollary 1.** $j = \arg \max(diag(\mathbb{E}[\frac{\partial s(x)}{\partial x}])) \Rightarrow x_j$'s sum of parent score $\sum_{i=0}^{d} \mathcal{P}_{i,j}$ is minimal $\Rightarrow x_j$ is a leaf node

The detailed proof is given in the Appendix A.5. Corollary 1 shows that Theorem 1 is actually finding the minimal sum of the parent score. Then, the node with the minimal total causal effect is a leaf node. This corollary implies the association between parent score and Theorem 1. Thus, the parent score can be considered as a unified metric during topological ordering and post-processing.

### 4.3 Pre-pruning and Edge Supplement

Previous research has demonstrated that redundant edges can be successfully eliminated through CAM pruning, which applies significance testing of covariates based on generalized additive models. This technique is widely used in many strong baselines [11, 23, 28]. However, CAM pruning is time-consuming and only utilizes the topological ordering information. The proposed metric, parent score, can further provide more information on causal effects. Thus, CaPS introduces the pre-pruning and edge supplement operations before/after CAM pruning process to accelerate pruning by removing edges with low parent score and restore removed edges with strong parent score.

**Pre-pruning.** Before CAM pruning, we use low-confidence parents to pre-prune the initial graph, which can remove the low-confidence edges and reduce the searching space for CAM pruning. For each node, we use the maximum value of their parents to determine the threshold for pre-pruning. Specifically, for any $x_i, x_j \in \mathbf{x}$, we mask the adjacency matrix $\mathcal{A}_{j,i} = 0$ if $\mathcal{P}_{i,j} < \frac{\max(\mathcal{P}_{i,:})}{\lambda}$, where $\lambda$ is a hyperparameter that represents the rigor in prepruning. This design can greatly speed up the pruning process, especially when the number of nodes is large (see Appendix C.5).

**Edge supplement.** After CAM pruning, we use high-confidence parents to supplement the edge, which can remedy errors in topological ordering and incorrect deletion in CAM pruning. With CAM pruning, the existing edges are likely to be the correct edges in a real causal graph. Thus, the parent score in current edges is used to automatically determine the threshold for edge supplement. For any $x_i, x_j \in \mathbf{x}$, we supplement the edge $\mathcal{A}_{j,i} = 1$ when the following conditions are satisfied. First, $\mathcal{P}_{i,j} > \lambda \cdot \operatorname{avg}(\mathcal{P}^\top \odot \mathcal{A})$, where $\odot$ denotes the Hadamard product, and $\operatorname{avg}(\cdot)$ returns the average value of a matrix. Here, we use the same rigor $\lambda$ for pre-pruning and edge supplement. Second, $\mathcal{A}$ is acyclic after supplementing the current edge. Note that edges added later may potentially violate

acyclicity, so we use a greedy strategy to prioritize adding the edge with a higher parent score. The pseudocode of post-processing are released in Appendix B.

## 4.4 Computational Complexity

For Algorithm 1, the computational complexity is mainly related to the estimation of score's Jacobian $d$ times, which is $\mathcal{O}(d \cdot n^3)$, with $n$ for the number of samples and $d$ for the feature dimension. For post-processing, the computational complexity is $\mathcal{O}(d^2 + d \cdot r^{(n,m)} + s \cdot (d + e + s))$, which can be considered as two steps. For the pruning step, the computational complexity of the original CAM pruning is $\mathcal{O}(d \cdot r^{(n,d)})$, where $r^{(n,d)}$ is the complexity function of training a generalized additive model. With pre-pruning, the computational complexity of pruning can be reduced to $\mathcal{O}(d^2 + d \cdot r^{(n,m)})$, where $m \leq d$ is the maximum number of parents for each node. For the edge supplement step, the computational complexity is $\mathcal{O}(s \cdot (d + e + s))$ due to acyclic testing, where $s$ denotes the number of candidate edges and $e < \frac{d \cdot (d-1)}{2}$ denotes the number of edges remaining after pruning in a DAG. Since we only want to supplement the edges with high confidence, $s$ tends to be a small value in the implementation. Despite this cubic complexity of $n$, the actual-time growth is close to linear since many CaPS operations are GPU-friendly, which is detailed in Appendix C.5.

## 5 Experiments

### 5.1 Baselines and Settings

**Baselines.** This paper benchmarks CaPS against eight strong state-of-the-art baselines designed for:

*Linear:* NOTEARS [15] and GOLEM [10], two strong linear methods with continuous optimization.

*Nonlinear:* GraNDAG [28] formulates neural network paths and a connectivity matrix, and substitutes them into the acyclicity penalty; Five ordering-based methods (CAM, VI-DP-DAG, SCORE, DiffAN, and DAGuerreotype) are chosen for comparison. DAGuerreotype has two versions: a linear one (DAGuerreotype-L) and a nonlinear one (DAGuerreotype-N). Both variants are evaluated in the synthetic data experiments to ensure a fair comparison.

**Metrics.** Three metrics in causal discovery are adopted for evaluation: the structural Hamming distance (SHD), the structural intervention distance (SID) [29], and the F1 score. SHD evaluates the number of edges that must be altered to make the estimated causal graph match the true causal graph. SID assesses the number of interventional distributions in the true causal graph that are disrupted in the estimated causal graph. Lower values for SHD and SID are desirable. SHD favors sparser estimated causal graphs, whereas SID favors denser estimated causal graphs. Therefore, doing well in only one of these two metrics does not necessarily mean effectiveness. F1 score measures the balance between precision and recall, with higher values indicating better performance.

**Settings.** We used the settings from the respective papers for all the baselines. For some methods that have multiple versions, such as GOLEM and DAGuerreotype, we reported the results of the version that gave the best performance on the corresponding dataset. The only hyperparameter of CaPS was rigor $\lambda$, which we set to $\lambda = 50$ for all datasets to avoid any dataset-specific tuning.

**Datasets.** Synthetic data are created using the Erdös-Rényi (ER) [30] or Scale-Free (SF) models[31] with different linear and nonlinear proportion. We set the number of nodes $d = 10$ and the number of samples $n = 2000$ by default, while $d = 20, 50$ and $n = 1000, 5000$ are also given. Real dataset contains a protein expression dataset *Sachs* [1] and a pseudoreal transport network dataset *Syntern* [32]. Details and more insights of the synthetic and the real data can be found in Appendix C.1.

### 5.2 Synthetic Data

Fig.2 shows the experiment results of eight baselines. We can observe that CaPS performs better for both sparser (SynER1) and denser (SynER4) graphs in almost all ranges, especially when the linear proportions are greater than 0.25. We also note that GOLEM's performance decreases with increasing nonlinear proportion, and SCORE's performance decreases with increasing linear proportion, which could be due to their assumptions not being met. In contrast, CaPS performs consistently well in almost all ratios. Experiments with the SynSF1 and SynSF4 datasets show similar results, and further information can be found in Appendix C.2.

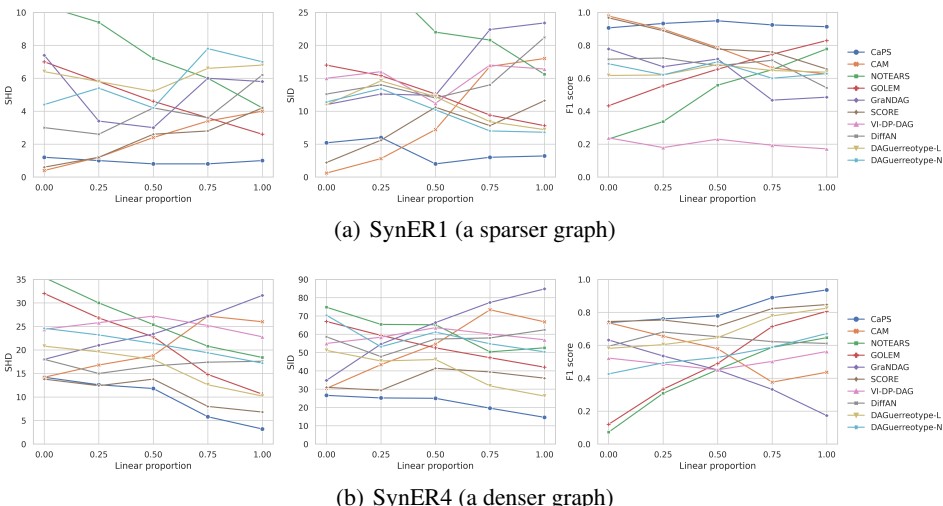

(a) SynER1 (a sparser graph)

(b) SynER4 (a denser graph)

Figure 2: Results of SynER1 and SynER4 with different linear proportions, where linear proportion equal to 0.0 means all relations are nonlinear and 1.0 means all relations are linear.

## 5.3 Real Data

The results of the real data are presented in Table 1. CaPS achieves the highest SHD and F1 scores on Sachs, with SID coming second to VI-DP-DAG. VI-DP-DAG had the best SID but the worst SHD, as it discovers a large number of false edges. On Syntren, GraNDAG is the top performer since the pattern of this dataset is not friendly to ordering-based methods (see Appendix C.1). However, CaPS achieves the best performance compared to other ordering-based methods.

To investigate the impact of each component of CaPS, we replace Theorem 1 with a random topological ordering (w/o Theorem 1) or turn off the pre-pruning and edge supplement with parent score (w/o Parent Score). The results show that Theorem 1 makes a major contribution to CaPS, and the parent score can further improve it. Actually, the performance of CaPS can be further improved by adjusting its hyperparameter $\lambda$. The sensitivity of different $\lambda$ is further analyzed in Appendix C.3.

Table 1: Results of real-world datasets, including three methods based on acyclicity constraint and five ordering-based methods. More baselines are given in Appendix C.4.

| Dataset | Sachs | | | Syntren | | |
|---|---|---|---|---|---|---|
| Metrics | SHD↓ | SID↓ | F1↑ | SHD↓ | SID↓ | F1↑ |
| NOTEARS | 12.0±0.00 | 46.0±0.00 | 0.387±0.000 | 33.9±4.57 | 192.8±54.73 | 0.164±0.085 |
| GOLEM | 17.0±0.00 | 44.0±0.00 | 0.421±0.000 | 43.7±10.72 | 177.4±56.55 | 0.163±0.066 |
| GraNDAG | 13.2±0.75 | 54.0±1.10 | 0.373±0.064 | **26.5±6.45** | 155.3±58.11 | **0.344±0.104** |
| CAM | 12.0±0.00 | 55.0±0.00 | 0.444±0.000 | 38.0±5.59 | 178.6±44.56 | 0.223±0.099 |
| VI-DP-DAG | 42.6±1.36 | **40.0±5.66** | 0.340±0.037 | 182.6±4.29 | **144.3±35.00** | 0.069±0.039 |
| SCORE | 12.0±0.00 | 45.0±0.00 | 0.444±0.000 | 37.5±4.20 | 197.1±63.71 | 0.183±0.091 |
| DiffAN | 12.2±0.98 | 46.2±6.18 | 0.434±0.078 | 44.1±8.29 | 188.7±55.16 | 0.191±0.095 |
| DAGuerreotype | 17.9±0.54 | 51.4±0.49 | 0.118±0.034 | 87.9±9.60 | 157.7±48.90 | 0.125±0.047 |
| CaPS | **11.0±0.00** | 42.0±0.00 | **0.500±0.000** | 37.2±5.04 | 178.9±55.58 | 0.230±0.072 |
| w/o Theorem 1 | 17.0±3.50 | 54.0±3.40 | 0.257±0.061 | 51.6±8.82 | 180.0±66.80 | 0.218±0.090 |
| w/o Parent Score | 12.0±0.00 | 45.0±0.00 | 0.444±0.000 | 34.8±3.37 | 188.0±57.58 | 0.222±0.083 |

## 5.4 Analysis Experiments

**Larger-scale datasets & actual-time cost.** Despite cubic complexity in samples size $n$, the bottleneck of actual-time growth often lies in the number of nodes $d$ in causal graph since many $n$-related operations are GPU-friendly. In Fig. 3, we illustrate the performance with actual-time cost for all baselines in larger-scale SynER1 ($d = 20$ and $d = 50$) with 0.5 linear proportion. CaPS consistently achieves best performance in larger-scale causal graph while its time cost is competitive. The full

experimental results with actual training time of larger-scale causal graph ($d = 20$ and $d = 50$) and different samples size ($n = 1000$ and $n = 5000$) can be found in Appendix C.5.

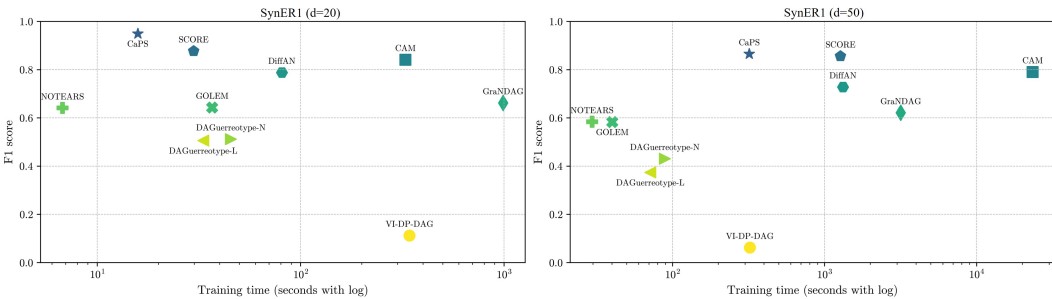

Figure 3: F1 score and training time of SynER1 with larger-scale causal graph.

**Order divergence.** Reisach et.al. [33] cautioned that some synthetic datasets may be so simple that sorting with minimal variance can be successful. To demonstrate the efficacy of CaPS, a new metric called "order divergence" [11] was introduced for evaluation, along with a new baseline *sortnregress* which orders nodes by increasing marginal variance. The results demonstrates that CaPS has a much better order divergence than *sortnregress*, indicating that variance is not a reliable indicator of the topological ordering in our synthetic datasets. CaPS consistently has the best or a competitive order divergence in different datesets. Details can be found in Appendix C.6.

**Beyond our assumptions.** We also explore the performance of CaPS under other settings of noise. The results show that CaPS can be effective in situations that go beyond our assumed conditions, which suggests that our approach has the potential to be applied in various other scenarios, and it is possible to consider loosening the assumptions in the future. Details can be found in Appendix C.7.

**Case visualization.** Fig. 4 shows the another advantage of CaPS. Compared to the second best baseline, CaPS performs better under all metrics while it provides more information on causal effects. The parent score captures most of the ground-truth edges and the estimated weights are similar to the actual values, indicating that the parent score accurately reflects the strength of causal effect.

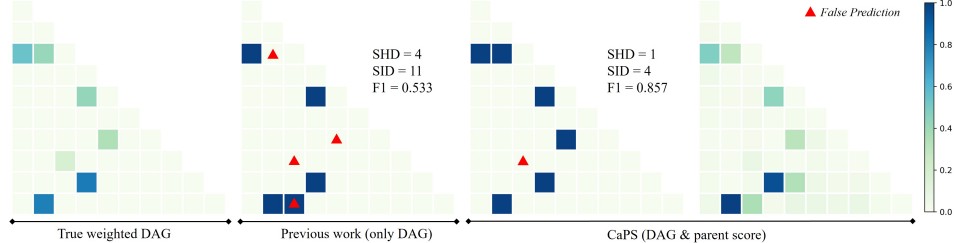

Figure 4: Visualization on SynER1 dataset. Darker colors indicate stronger causal effects.

## 6 Conclusion

This paper introduces CaPS that is capable of handling datasets with linear and nonlinear relations, which is a common occurrence in real-world applications. We propose a novel identification criterion for topological ordering for both types of relation, as well as a new metric, "parent score", to measure the strength of the average causal effect and used for edge removal and supplementation. Our solutions have been tested on synthetic data with varying linear and nonlinear relationship ratios and have been found to be more effective than existing order-based work and state-of-the-art baselines.

There remain two interesting directions to be explored in future work. (1) Since the experimental results have been encouraging in some cases beyond our assumption, we are striving to broaden the identifiability conditions to more relaxed conditions. (2) The new metric, "parent score", is likely to have more application scenarios. It is possible to apply as a plug-and-play information of causality.

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

# Appendix

## A Theoretical Analysis

### A.1 Reviewing LISTEN

To show why LISTEN does not work under nonlinear causal relations, we review the derivation of precision matrix under linear relations. LISTEN uses the linear SEM $X = \mathbf{B}X + N$, where $\mathbf{B}$ is the autoregression matrix, and $N = (\epsilon_1, ..., \epsilon_d)$ is independent Gaussian noises. Since $B$ is a weight matrix of a DAG, $(\mathbf{I} - \mathbf{B})$ is invertible. Then, the covariance matrix of X is equivalent to

$$\mathbf{\Sigma} = \mathbb{E}[XX^\top] = \mathbb{E}[(\mathbf{I} - \mathbf{B})^{-1}NN^\top((\mathbf{I} - \mathbf{B})^{-1})^\top] = (\mathbf{I} - \mathbf{B})^{-1}\mathbf{D}((\mathbf{I} - \mathbf{B})^{-1})^\top$$

where $\mathbf{D} = \mathbf{Diag}(\sigma_1, ..., \sigma_d)$ is the covariance matrix of Gaussian noise. The precision matrix $\mathbf{\Omega}$ is the inverse covariance matrix, where $\mathbf{\Omega} = (\mathbf{I} - \mathbf{B})^\top\mathbf{D}^{-1}(\mathbf{I} - \mathbf{B})$. When the causal relations is nonlinear, we cannot write it directly in the form of a linear weight matrix $\mathbf{B}$ multiplied by $X$. Thus, the connection between the precision matrix and the adjacency matrix of the true causal graph no longer holds under nonlinear causal relations. And we can no directly use the precision matrix to determine the leaf nodes.

### A.2 The Proof of Theorem 1

**Theorem 2.** *Let $s(x) = \nabla \log p(x)$ be the score and let $diag(\cdot)$ be the diagonal elements of the matrix. For any $x_j$ in the causal graph $\mathcal{G}$:*

$$j = \arg\max(diag(\mathbb{E}[\tfrac{\partial s(x)}{\partial x}])) \Rightarrow x_j \text{ is a leaf node}$$

*Proof.* For an arbitrary node $x_j$ in the causal graph $\mathcal{G}$, we focus on the diagonal of the score's Jacobian. The expression of $\frac{\partial s_j(x)}{\partial x_j}$ is derived as follows:

$$\frac{\partial s_j(x)}{\partial x_j} = -\frac{1}{\sigma_j^2} - \sum_{i \in ch(j)} \frac{1}{\sigma_i^2}(\frac{\partial f_i}{\partial x_j}(pa_i(x)))^2 + \sum_{i \in ch(j)} \frac{\partial^2 f_i}{\partial x_j^2}(pa_i(x)) \cdot \frac{x_i - f_i(pa_i(x))}{\sigma_i^2} \quad (10)$$

The residual $x_i - f_i(pa_i(x))$ in the last term of the RHS of Eq.10 is additive noise $\epsilon_i$ as stated in Equation 1, which implies that $\frac{x_i - f_i(pa_i(x))}{\sigma_i^2} = \frac{\epsilon_i}{\sigma_i^2} \sim \mathcal{N}(0, \frac{1}{\sigma_i^2})$. Furthermore, since $\epsilon_i$ and $pa_i(x)$ are independent of each other in our SEM, the expectation of the last term can be expressed as:

$$\mathbb{E}[\sum_{i \in ch(j)} \frac{\partial^2 f_i}{\partial x_j^2}(pa_i(x)) \cdot \frac{x_i - f_i(pa_i(x))}{\sigma_i^2}]$$
$$= \mathbb{E}[\sum_{i \in ch(j)} \frac{\partial^2 f_i}{\partial x_j^2}(pa_i(x))] \cdot \mathbb{E}[\frac{\epsilon_i}{\sigma_i^2}] = 0 \quad (11)$$

With Eq.10 and Eq.11, the expectation of $\frac{\partial s_j(x)}{\partial x_j}$ can be restated as:

$$\mathbb{E}[\frac{\partial s_j(x)}{\partial x_j}] = -\frac{1}{\sigma_j^2} - \sum_{i \in Ch(j)} \frac{1}{\sigma_i^2}\mathbb{E}[(\frac{\partial f_i}{\partial x_j}(pa_i(x)))^2] \quad (12)$$

According to Eq.12, the expectation of $\frac{\partial s_j(x)}{\partial x_j}$ is only dependent on the current node $x_j$ and its children. Suppose that $x_l$ is a leaf node and $x_n$ is a non-leaf node, we have $\mathbb{E}[\frac{\partial s_l(x)}{\partial x_l}] = -\frac{1}{\sigma_l^2}$ and $\mathbb{E}[\frac{\partial s_n(x)}{\partial x_n}] = -\frac{1}{\sigma_n^2} - \sum_{i \in Ch(n)} \frac{1}{\sigma_i^2}\mathbb{E}[(\frac{\partial f_i}{\partial x_n}(pa_i(x)))^2]$.

(i) If the sufficient condition $(i)$ is satisfied, we have non-decreasing variance of noises. For any two noises $\epsilon_i$ and $\epsilon_j$ of different nodes, $\sigma_j \geq \sigma_i$ if $\pi(i) < \pi(j)$. Since $x_l$ is a leaf node

and $x_n$ is a non-leaf node, $\pi(n) < \pi(l)$, we have $\sigma_l \geq \sigma_n$. And, $\frac{1}{\sigma_i^2}\mathbb{E}[(\frac{\partial f_i}{\partial x_j}(pa_i(x)))^2] \geq 0$. Therefore, we can get

$$\mathbb{E}[\frac{\partial s_l(x)}{\partial x_l}] = -\frac{1}{\sigma_l^2} \geq -\frac{1}{\sigma_n^2} \geq -\frac{1}{\sigma_n^2} - \sum_{i \in Ch(n)} \frac{1}{\sigma_i^2}\mathbb{E}[(\frac{\partial f_i}{\partial x_n}(pa_i(x)))^2] = \mathbb{E}[\frac{\partial s_n(x)}{\partial x_n}] \quad (13)$$

This equation implies that the value of the leaf node will always be greater than the non-leaf node in the diag of the expectation of score's Jacobian. And, for any causal graph, there is at least one leaf node $x_l$. Thus, the $\arg\max(\text{diag}(\mathbb{E}[\frac{\partial s(x)}{\partial x}]))$ is the index of the leaf node, i.e., $j = \arg\max(\text{diag}(\mathbb{E}[\frac{\partial s(x)}{\partial x}])) \Rightarrow x_j$ is a leaf node.

(ii) If the sufficient condition $(ii)$ is satisfied, we have non-weak causal effect of parents. For any non-leaf nodes $x_j$, $\sum_{i \in Ch(j)} \frac{1}{\sigma_i^2}\mathbb{E}[(\frac{\partial f_i}{\partial x_j}(pa_i(x)))^2] \geq \frac{1}{\sigma_{\min}} - \frac{1}{\sigma_j}$, where $\sigma_{\min}$ is the minimum variance for all noises. Thus, the following inequality holds.

$$\begin{aligned}\mathbb{E}[\frac{\partial s_l(x)}{\partial x_l}] = -\frac{1}{\sigma_l^2} &\geq -\frac{1}{\sigma_{\min}^2} = -\frac{1}{\sigma_n^2} - (\frac{1}{\sigma_{\min}^2} - \frac{1}{\sigma_n^2}) \\ &\geq -\frac{1}{\sigma_n^2} - \sum_{i \in Ch(n)} \frac{1}{\sigma_i^2}\mathbb{E}[(\frac{\partial f_i}{\partial x_n}(pa_i(x)))^2] = \mathbb{E}[\frac{\partial s_n(x)}{\partial x_n}]\end{aligned} \quad (14)$$

Therefore, it can be obtained in the same way that $j = \arg\max(\text{diag}(\mathbb{E}[\frac{\partial s(x)}{\partial x}])) \Rightarrow x_j$ is a leaf node.

Thus, under the sufficient conditions $(i)$ or $(ii)$, we can proof that the node in $\arg\max(\text{diag}(\mathbb{E}[\frac{\partial s(x)}{\partial x}]))$ will always be the leaf node. $\qquad\square$

### A.3 Derivation of SATE and parent score

To illustrate the meaning of this definition, we propose a new metric of the Squared Average Treatment Effect (SATE) extended from Average Treatment Effect (ATE, $\mathbb{E}[Y^{(T=1)} - Y^{(T=0)}]$), which can be rewritten as

$$\text{SATE}_i^j = \mathbb{E}[(x_i^{(T_j=1)} - x_i^{(T_j=0)})^2] \quad (15)$$

In order to establish an association between the parent score and the causal effect, the treatment on $x_j$ is defined as follows:

$$\begin{aligned}x_i^{(T_j=1)} &= f_i(pa_i(x + \Delta_j)) + \epsilon_i \\ x_i^{(T_j=0)} &= f_i(pa_i(x)) + \epsilon_i\end{aligned} \quad (16)$$

where $\Delta_j = [0, ..., \delta, ..., 0]$ is a one-hot vector denoting an additive treatment on $x_j$. Here, for each $j$, we design the $j$-th value of $\Delta_j$ as $\delta$ which denotes the strength of treatment, which is a very small positive value. Due to the small value of $\delta \to 0^+$, for each variable $x_i$, we can use the optimal linear approximation to represent $x_i^{(T_j=1)}$. Then, for the $j$-th treatment, SATE is equal to:

$$\text{SATE}_i^j = \delta^2 \cdot \mathbb{E}[(\frac{\partial f_i}{\partial x_j}(pa_i(x)))^2] \quad (17)$$

For each treatment, we treat with the same strength $\delta$. Therefore, the average causal effect can be represented by $\mathbb{E}[(\frac{\partial f_i}{\partial x_j}(pa_i(x)))^2]$. Obviously, its value is zero when $x_j$ is not a parent of $x_i$. If $x_j$ is a parent of $x_i$, $\mathbb{E}[(\frac{\partial f_i}{\partial x_j}(pa_i(x)))^2]$ indicates the influence strength from $x_j$ to $x_i$. Thus, scaled by the variance of the children's noise, we can use $\frac{1}{\sigma_i^2}\mathbb{E}[(\frac{\partial f_i}{\partial x_j}(pa_i(x)))^2]$ to approximately represent the strength of the average causal effect, which is called the parent score. This new metric considers both causal effect and noise variance, which can be considered as the visible part of causal effect if the variance of noise is close or equal. This assumption does not affect the sufficient conditions for the identifiability of topological ordering and is only supposed in post-processing, which has been used frequently in previous works [8, 13].

### A.4 Derivation of computing parent score

Parent score cannot be obtained directly from the summation of average causal effects of a node on its children in Eq.12, thus we propose an iterative decoupling process and define

$$\mathcal{J} = \{\mathbb{E}[\frac{\partial s_1(x)}{\partial x_1}], \mathbb{E}[\frac{\partial s_2(x)}{\partial x_2}], ..., \mathbb{E}[\frac{\partial s_d(x)}{\partial x_d}]\} \tag{18}$$

to denote the expectation of the diagonal of the score's Jacobian. By removing the node $x_i$, a new vector $\mathcal{J}_{-i}$ is defined as follows:

$$\mathcal{J}_{-i} = \{\mathbb{E}[\frac{\partial s_1(x_{-i})}{\partial x_1}], \mathbb{E}[\frac{\partial s_2(x_{-i})}{\partial x_2}], ..., \mathbb{E}[\frac{\partial s_i(x)}{\partial x_i}], ..., \mathbb{E}[\frac{\partial s_d(x_{-i})}{\partial x_d}]\} \tag{19}$$

where $x_{-i}$ represents the remaining data after removing the feature of $i$-th dimension. For the $i$-th element of $\mathcal{J}_{-i}$, we fill the $i$-th element of $\mathcal{J}$. To simplify the notation, we define in this subsection that $\mathcal{J}^{(j)}$ and $= \mathcal{J}_{-i}^{(j)}$ are the $j$-th element of the corresponding vector. Then, for $j$-th element, we have

$$\mathcal{J}^{(j)} = -\frac{1}{\sigma_j^2} - \sum_{k \in Ch(j)} \frac{1}{\sigma_k^2} \mathbb{E}[(\frac{\partial f_k}{\partial x_j}(pa_k(x)))^2] \tag{20}$$

$$\mathcal{J}_{-i}^{(j)} = -\frac{1}{\sigma_j^2} - \sum_{k \in Ch(j)/x_i} \frac{1}{\sigma_k^2} \mathbb{E}[(\frac{\partial f_k}{\partial x_j}(pa_k(x)))^2] \tag{21}$$

where $Ch(j)/x_i$ denotes the set which the element $x_i$ has been removed if it is the children of $x_j$. If $x_i$ is not the children of $x_j$, $Ch(j)$ is equal to $Ch(j)/x_i$. According to the definition of parent score, we have

$$\mathcal{P}_{i,j} = \mathcal{J}_{-i}^{(j)} - \mathcal{J}^{(j)} = \begin{cases} \frac{1}{\sigma_i^2}\mathbb{E}[(\frac{\partial f_i}{\partial x_j}(pa_i(x)))^2], & x_j \in pa_i(x) \\ 0, x_j \notin pa_i(x) \end{cases} \tag{22}$$

Then, each row vector of the matrix of parent score $\mathcal{P} \in \mathbb{R}^{d \times d}$ is equivalent to:

$$\mathcal{P}_{i,:} = \mathcal{J}_{-i} - \mathcal{J} \tag{23}$$

Thus, we can obtain the parent score by iteratively removing nodes and estimating the score's Jacobian.

### A.5 The Proof of Corollary 1

**Corollary 1.** $j = \arg\max(diag(\mathbb{E}[\frac{\partial s(x)}{\partial x}])) \Rightarrow x_j$'s sum of parent score $\sum_{i=0}^d \mathcal{P}_{i,j}$ is minimal $\Rightarrow x_j$ is a leaf node

*Proof.* We first prove that $j = \arg\max(diag(\mathbb{E}[\frac{\partial s(x)}{\partial x}])) \Rightarrow x_j$'s sum of parent score $\sum_{i=0}^d \mathcal{P}_{i,j}$ is minimal. Under the sufficient condition $(ii)$, the sum of parent score of non-leaf node has a low bound $\frac{1}{\sigma_{\min}} - \frac{1}{\sigma_j}$. According to the definition of parent score, for any node $x_j$, we have $\sum_{i=0}^d \mathcal{P}_{i,j} = 0$ or $\sum_{i=0}^d \mathcal{P}_{i,j} \geq \frac{1}{\sigma_{\min}} - \frac{1}{\sigma_j}$. Thus, for an arbitrary element in the diagonal score's Jacobian, we have

$$\begin{cases} \mathbb{E}[\frac{\partial s_j(x)}{\partial x_j}] = -\frac{1}{\sigma_j^2}, & \sum_{i=0}^d \mathcal{P}_{i,j} = 0 \\ \mathbb{E}[\frac{\partial s_j(x)}{\partial x_j}] \leq -\frac{1}{\sigma_{\min}^2}, & \sum_{i=0}^d \mathcal{P}_{i,j} \geq \frac{1}{\sigma_{\min}} - \frac{1}{\sigma_j} \end{cases} \tag{24}$$

Obviously, $-\frac{1}{\sigma_j^2}$ is always greater than $-\frac{1}{\sigma_{\min}^2}$. Therefore, given a $x_j$ with $j = \arg\max(diag(\mathbb{E}[\frac{\partial s(x)}{\partial x}]))$, its sum of parent score is minimal and $\sum_{i=0}^d \mathcal{P}_{i,j} = 0$.

Then, we prove that $x_j$'s sum of parent score $\sum_{i=0}^d \mathcal{P}_{i,j}$ is minimal $\Rightarrow x_j$ is a leaf node. Suppose that $x_l$ is a leaf node and $x_n$ is a non-leaf node, according to the definition of parent score, we have $\sum_{i=0}^d \mathcal{P}_{i,n} > 0 = \sum_{i=0}^d \mathcal{P}_{i,l}$. Therefore, the node with the minimal sum of the parent score is a leaf node. $\qed$

# B    Pseudocode of Post-processing

CaPS introduces the pre-pruning and edge supplement process in post-processing, which considers more information about causal effect by using the parent score. The complete post-processing procedure is shown in Algorithm 2, where $\mathcal{P}_{\max}$ is the matrix broadcasted with the maximum value of each row of $\mathcal{P}$. For hyperparameter $\lambda$ selection, we set to $\lambda = 50$ for all datasets to avoid any dataset-specific tuning, which can be further optimized in appendix C.3.

---

**Algorithm 2** Post-processing

---

**Input**: data matirx $X \in \mathbb{R}^{n \times d}$, rigor $\lambda$, permutation $\pi$, parent score $\mathcal{P}$
**Output**: adjacency matrix $\mathcal{A}$

1:  initialize $\mathcal{A}$ using $\pi$
2:  $\mathcal{A} \leftarrow \mathcal{A} \odot \text{int}(\mathcal{P} < \frac{\mathcal{P}_{\max}}{\lambda})^{\top}$ // pre-pruning
3:  $\mathcal{A} \leftarrow$ CAM pruning($\mathcal{A}$, $X$)
4:  $\mathcal{E} \leftarrow \text{int}(\mathcal{P} > \lambda \cdot \text{avg}(\mathcal{P}^{\top} \odot \mathcal{A}))$
5:  **for** each edge (i, j) in $\mathcal{E}$ sorted by $\mathcal{P}_{i,j}$ **do**
6:      $\mathcal{A}_{j,i} \leftarrow 1$ if $\mathcal{A}$ is still acyclic // edge supplement
7:  **end for**
8:  **return** $\mathcal{A}$

---

# C    Experiments

All experiments were run on EPYC 7552*2 with 512G memory and NVIDIA RTX 4090 32GB.

## C.1    Dataset Details

Synthetic data are created using the Erdös-Rényi (ER) [30] or Scale-Free (SF) models[31]. The degree of each node in the ER graph is relatively even, whereas some nodes in the SF graph may have very high degrees. Each dataset has 2000 samples and 10 nodes by default, while larger-scale datasets and different samples size are also given in Appendix C.5. The sparsity of the graph is altered by changing the average number of edges to either $d$ or four times $d$, referred to as SynER1 and SynER4. The weight of each edge is randomly chosen from the range of $[-1, -0.1] \cup [0.1, 1]$. To create datasets with both linear and nonlinear relationships between variables, we set the noise distribution to be Gaussian with a mean of 0 and a variance of 1. To generate linear causal functions $f_i$, we use weighted causal graphs. For nonlinear causal functions $f_i$, we sample Gaussian processes with a unit bandwidth RBF kernel and multiply them by the weights in the causal graph, similar to SCORE and DiffAN. A collection of five datasets is created for SynER1 and SynER4, with a range of linear and nonlinear causal connections, from 0.0 (all nonlinear) to 1.0 (all linear) with step 0.25.

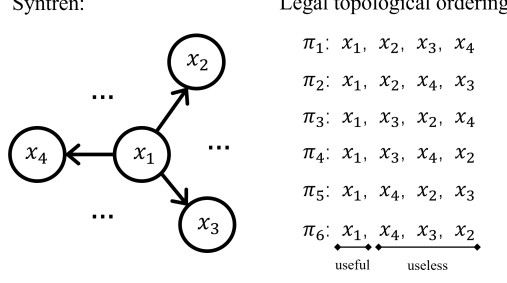

Figure 5: Example of the Syntren dataset.

Two real datasets are adopted. Sachs [1], a protein signaling network based on protein expression levels and phospholipids, consists of 11 nodes, 853 observations, and 17 edges from ground-truth causal graph; Syntren [32], a pseudoreal data set sampled from the Syntren generator, consists of 10 transcriptional networks, each consisting of 500 observations and a DAG consisting of $d = 20$ nodes and edges with $e \in \{20, ..., 25\}$. For the two real datasets, we only have ground-truth DAG but no relation types. To quantitatively give more insight into their linear and nonlinear proportion, we estimate the type of relation in the DAG with a rough metric: Pearson correlation$>$0.5 as linear

relation. Accordingly, Sachs has 0.18 linear ratio while Syntren averaged 0.92 for different DAGs ($0.72 \sim 1.0$). Although these ratios are rather rough, it is evident that mix relations are widespread in realistic applications. This explains why CaPS is effective in different datasets.

Fig. 5 shows an example of Syntren dataset, which is a special dataset containing many star networks. In this network structure, only the topological ordering of the first node $x_1$ is meaningful, and the other nodes can be ordered randomly after $x_1$. This star structure is not friendly to ordering-based methods, since topological ordering provides more limited information in such a causal graph. However, among all ordering-based methods, CaPS achieves the highest performance in this dataset.

## C.2  Additional Synthetic Datasets

Fig. 6 shows the experiments results of the SynSF1 and SynSF4 datasets. We can observe that CaPS performs better for both sparser (SynSF1) and denser (SynSF4) graphs in almost all ranges, especially when the linear proportions above 0.25. The results show that CaPS performs consistently well in almost all ratios and different DAG generator (ER or SF).

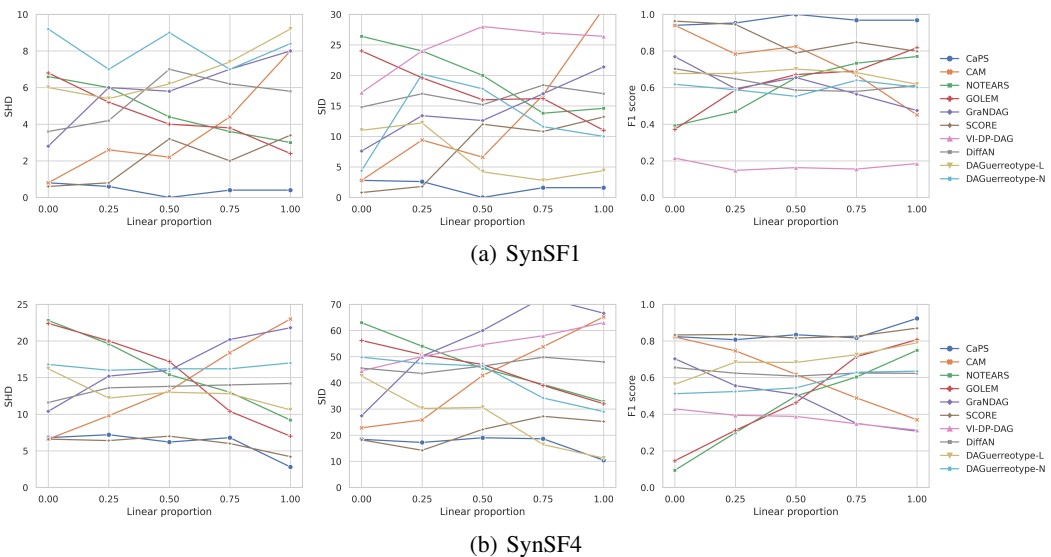

Figure 6: Results of SynSF1 and SynSF4 with different linear proportions.

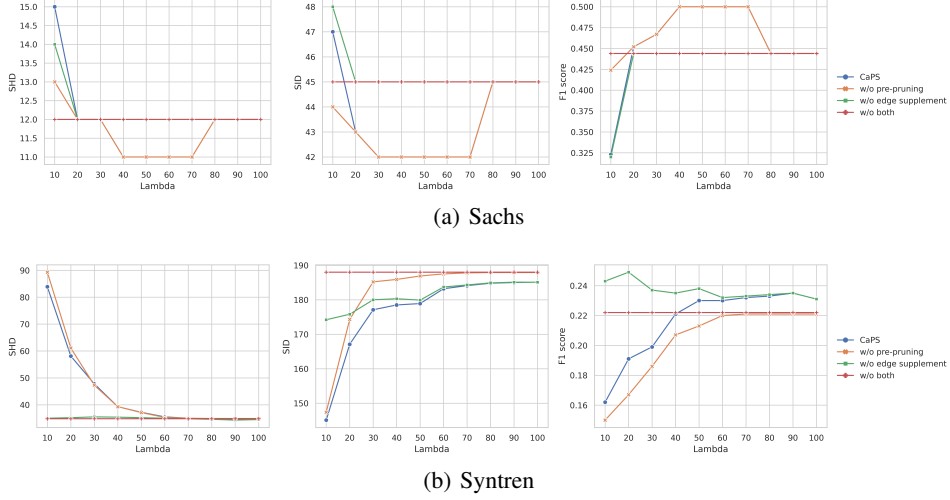

Figure 7: Performance under different hyperparameter $\lambda$.

## C.3 Hyperparameter Analysis

We further study in Fig. 7 the effect of different $\lambda$ on the performance of pre-pruning and edge supplement on the real datasets. As we mentioned in the ablation study, edge supplement is more effective on the Sachs dataset, while pre-pruning is more effective on the Syntren dataset. For Sachs dataset, edge supplement works best under $\lambda \in [40, 70]$, and CaPS can maintain better or competitive performance in a large range of the hyperparameter $\lambda$. For Syntren dataset, since it is a difficult dataset for causal discovery, it could be possible to add some incorrect edges when $\lambda$ is too small. Pre-pruning shows more potential in the Syntern dataset, which can be further improved by selecting different hyperparameter $\lambda$. Based on the results shown in Fig. 7, we can see that the performance of CaPS can be further improved by setting different hyperparameters for pre-pruning and edge supplement, e.g., $\lambda = 20$ for pre-pruning and $\lambda = 70$ for edge supplement.

## C.4 More Baselines

We additionally consider six baselines of continuous optimization: DAG-GNN [16], a nonlinear extension of NOTEARS that uses an evidence lower bound as score; GAE [17], an autoencoder based method that further extends the NOTEARS and DAG-GNN to facilitate nonlinear relations; NOTEARS-MLP [18] is a nonlinear extension of NOTEARS; DAGMA [19] uses a log-determinant acyclicity characterization; TOPO [20] optimizes with iteratively swapping pairs of nodes within the topological ordering; CASTLE [34] uses causal discovery as an auxiliary task to the prediction task. The results of the real data experiments on fourteen baselines are presented in Table 2. On the Syntren datasets, NOTEARS-MLP achieves suboptimal SID but poor SHD, and CASTLE achieves suboptimal SHD but poor SID, suggesting that their predictions are too dense or sparse. Our method achieved the highest SHD and F1 scores on Sachs, with SID second to VI-DP-DAG. VI-DP-DAG had the best SID but the worst SHD, as it discovered a large number of false edges.

Table 2: Results of real-world datasets.

| Dataset | Sachs | | | Syntren | | |
|---|---|---|---|---|---|---|
| Metrics | SHD↓ | SID↓ | F1↑ | SHD↓ | SID↓ | F1↑ |
| CAM | 12.0±0.00 | 55.0±0.00 | 0.444±0.000 | 38.0±5.59 | 178.6±44.56 | 0.223±0.099 |
| NOTEARS | 12.0±0.00 | 46.0±0.00 | 0.387±0.000 | 33.9±4.57 | 192.8±54.73 | 0.164±0.085 |
| DAG-GNN | 14.0±0.00 | 45.0±0.00 | 0.397±0.005 | 32.4±4.86 | 191.2±52.66 | 0.156±0.066 |
| GAE | 17.2±1.60 | 50.2±2.93 | 0.119±0.112 | 79.7±13.13 | 156.3±79.10 | 0.127±0.077 |
| NOTEARS-MLP | 14.4±0.49 | 46.0±0.00 | 0.359±0.005 | 114.7±30.50 | 150.4±48.51 | 0.093±0.037 |
| DAGMA | 13.0±0.00 | 46.0±0.00 | 0.370±0.000 | 35.6±6.70 | 191.8±54.50 | 0.186±0.062 |
| TOPO | 21.6±0.40 | 44.0±0.00 | 0.303±0.000 | 39.0±12.90 | 191.0±57.20 | 0.227±0.118 |
| GOLEM | 17.0±0.00 | 44.0±0.00 | 0.421±0.000 | 43.7±10.72 | 177.4±56.55 | 0.163±0.066 |
| CASTLE | 15.8±4.12 | 46.8±0.98 | 0.299±0.053 | 30.5±8.14 | 206.1±61.23 | 0.096±0.081 |
| GraNDAG | 13.2±0.75 | 54.0±1.10 | 0.373±0.064 | **26.5±6.45** | 155.3±58.11 | **0.344±0.104** |
| VI-DP-DAG | 42.6±1.36 | **40.0±5.66** | 0.340±0.037 | 182.6±4.29 | **144.3±35.00** | 0.069±0.039 |
| SCORE | 12.0±0.00 | 45.0±0.00 | 0.444±0.000 | 37.5±4.20 | 197.1±63.71 | 0.183±0.091 |
| DiffAN | 12.2±0.98 | 46.2±6.18 | 0.434±0.078 | 44.1±8.29 | 188.7±55.16 | 0.191±0.095 |
| DAGuerreotype | 17.9±0.54 | 51.4±0.49 | 0.118±0.034 | 87.9±9.60 | 157.7±48.90 | 0.125±0.047 |
| CaPS | **11.0±0.00** | 42.0±0.00 | **0.500±0.000** | 37.2±5.04 | 178.9±55.58 | 0.230±0.072 |

## C.5 Larger-scale datasets & actual-time cost.

Table 3 shows the experimental results of large-scale datasets and the actual-time cost. Here, we varied the settings of the synthetic dataset SynER1 to be: $d = 20$, $d = 50$, $n = 1000$ and $n = 5000$. For better presentation, we abbreviate the linear proportion as prop. and the baseline DAGuerreotype as DAGu. On different scales of datasets, CaPS consistently achieves the best performance in most linear proportions, especially in mixed scenarios (prop.=0.25, 0.5, 0.75). Our method also achieves the best or competitive results even in pure linear and pure nonlinear scenarios compared to existing strong baselines. About the actual-time cost, despite cubic complexity in samples size $n$, the bottleneck of actual-time growth often lies in the number of nodes $d$ in causal graph since many $n$-related operations are GPU-friendly. However, since CaPS uses pre-pruning for acceleration in the post-processing stage, our method run with competitive actual-time cost even in large-scale datasets.

In terms of pruning time, most of the time is spent on CAM pruning. Pre-pruning can effectively remove the edges with low parent score, which can reduce the cost of fitting a generalized additive

Table 3: Results of larger-scale datasets with actual-time cost.

| Prop. | Metrics | NOTEARS | COLEM | GraNDAG | CAM | VI-DP-DAG | SCORE | DiffAN | DAGu.-L | DAGu.-N | CaPS |
|---|---|---|---|---|---|---|---|---|---|---|---|
| | | | | | **SynER1 (d=20)** | | | | | | |
| 0 | SHD | 17.4±1.7 | 15.2±1.8 | 5.3±1.2 | 1.4±0.8 | 123.2±4.9 | 1.4±0.8 | 6.2±3.7 | 23.6±2.0 | 18.3±2.6 | **0.8±0.4** |
| | SID | 81.2±16.1 | 81.0±22.6 | 18.6±7.7 | 8.4±6.1 | 33.4±7.3 | 5.4±3.2 | 31.6±17.5 | 63.3±23.5 | 46.6±22.8 | **3.4±2.8** |
| | F1 | 27.0±11.1 | 41.3±8.9 | 82.0±6.3 | 95.5±2.6 | 18.4±2.1 | 96.5±2.3 | 79.9±11.4 | 44.0±10.8 | 56.8±6.6 | **98.1±1.0** |
| 0.25 | SHD | 14.8±1.6 | 14.2±2.0 | 5.3±0.9 | 3.4±1.3 | 118.4±7.4 | **0.4±0.4** | 4.6±2.0 | 18.0±1.6 | 19.3±3.2 | 1.2±1.1 |
| | SID | 73.8±14.3 | 78.0±21.7 | 32.0±9.8 | 18.2±10.0 | 56.8±23.9 | **3.8±7.6** | 32.2±15.0 | 39.0±2.1 | 64.0±33.9 | 7.6±10.2 |
| | F1 | 44.1±5.7 | 43.9±12.7 | 79.8±6.0 | 87.1±5.4 | 16.4±3.1 | **98.6±1.8** | 81.9±8.5 | 61.5±3.7 | 52.6±10.2 | 96.0±3.9 |
| 0.5 | SHD | 10.8±1.7 | 10.0±1.6 | 9.6±2.6 | 5.4±2.6 | 125.4±8.3 | 4.6±3.9 | 6.6±2.6 | 25.3±6.6 | 27.0±5.0 | **4.2±3.0** |
| | SID | 56.6±13.3 | 53.6±16.9 | 63.3±21.6 | 21.0±9.5 | 85.2±20.0 | 17.2±13.3 | 24.2±12.5 | 72.3±37.0 | 40.6±9.7 | **17.0±11.4** |
| | F1 | 64.1±7.5 | 64.3±8.4 | 66.2±6.1 | 81.4±6.7 | 11.2±2.2 | 87.7±9.9 | 78.8±09.1 | 50.6±7.6 | 51.2±0.2 | **94.9±4.6** |
| 0.75 | SHD | 8.8±1.7 | 7.8±2.9 | 16.6±3.3 | 10.2±2.6 | 119.6±7.3 | 3.8±2.3 | 9.0±3.3 | 21.6±10.2 | 26.3±1.2 | **1.6±1.6** |
| | SID | 46.6±10.4 | 45.6±21.6 | 67.6±10.4 | 57.2±18.8 | 83.4±44.2 | 17.4±13.6 | 41.6±13.3 | 49.3±36.5 | 56.6±3.3 | **11.8±10.9** |
| | F1 | 72.5±4.3 | 7.14±11.0 | 43.4±13.6 | 63.1±10.7 | 13.8±4.9 | 88.2±4.4 | 70.9±6.7 | 56.1±16.9 | 48.5±2.0 | **94.9±4.6** |
| 1 | SHD | 7.0±0.8 | 4.4±1.4 | 18.3±0.4 | 12.8±2.9 | 119.2±8.2 | 6.2±2.0 | 9.8±1.7 | 27.0±8.2 | 26.3±0.9 | **2.0±2.2** |
| | SID | 41.8±12.2 | 24.0±2.2 | 74.6±6.5 | 60.2±20.0 | 84.6±35.4 | 30.6±22.6 | 34.8±5.8 | 51.6±25.7 | 64.6±6.7 | **12.4±12.1** |
| | F1 | 78.6±4.6 | 84.1±3.8 | 40.5±8.2 | 55.5±12.4 | 14.1±4.7 | 80.4±4.4 | 69.6±4.3 | 51.9±8.6 | 46.3±0.5 | **93.7±6.0** |
| Training time | | 6.7±0.6 | 36.7±0.7 | 990.1±93.9 | 327.7±5.7 | 343.1±110.6 | 29.8±1.2 | 80.9±1.0 | 33.2±2.6 | 45.4±2.6 | 15.8±3.3 |
| | | | | | **SynER1 (d=50)** | | | | | | |
| 0 | SHD | 43.0±4.8 | 39.8±4.7 | 26.6±7.7 | **5.2±2.8** | 795.2±29.5 | 6.6±3.0 | 17.0±3.4 | 96.3±13.9 | 56.3±6.2 | 7.2±4.4 |
| | SID | 281.2±114.3 | 270.4±94.0 | 173.3±77.6 | 25.2±7.7 | 198.6±79.5 | **24.8±13.2** | 105.8±55.7 | 262.0±137.3 | 168.3±50.0 | 56.6±49.0 |
| | F1 | 27.5±6.5 | 34.3±6.1 | 60.1±12.4 | **94.3±2.1** | 6.9±1.1 | 93.2±3.0 | 77.8±4.6 | 35.5±3.9 | 52.4±0.4 | 91.4±6.2 |
| 0.25 | SHD | 38.0±6.0 | 32.6±7.3 | 30.0±4.9 | 12.2±9.8 | 806.4±17.8 | 12.0±3.5 | 16.8±3.5 | 98.6±16.7 | 67.6±9.2 | **11.4±1.8** |
| | SID | 248.0±109.4 | 254.6±94.5 | 168.6±28.7 | **63.2±41.2** | 297.8±80.9 | 79.2±18.6 | 109.0±51.9 | 294.0±130.9 | 195.3±94.4 | 68.4±18.6 |
| | F1 | 40.7±10.6 | 48.5±11.4 | 58.4±10.2 | 84.7±10.5 | 5.7±5.1 | 84.4±5.4 | 76.8±4.0 | 32.5±3.6 | 47.5±2.4 | **85.7±3.3** |
| 0.5 | SHD | 29.8±4.3 | 27.4±1.9 | 25.6±6.9 | 16.4±7.9 | 816.6±35.5 | 12.0±5.5 | 20.8±4.7 | 94.0±10.6 | 80.3±9.5 | **11.4±4.4** |
| | SID | 201.4±113.8 | 186.6±57.8 | 143.0±8.8 | 76.0±46.6 | 231.6±75.7 | **65.2±48.2** | 134.8±82.8 | 285.3±175.8 | 218.6±117.7 | 72.8±47.5 |
| | F1 | 58.4±6.7 | 58.3±6.7 | 62.1±13.0 | 79.0±9.6 | 6.2±0.7 | 85.6±6.8 | 72.8±4.3 | 37.4±2.5 | 43.1±5.2 | **86.5±5.4** |
| 0.75 | SHD | 26.8±2.6 | 23.2±3.3 | 37.6±5.7 | 26.4±8.8 | 789.6±26.2 | 11.8±4.3 | 19.4±2.3 | 85.6±7.4 | 103.0±27.7 | **8.2±3.0** |
| | SID | 190.6±86.4 | 190.8±77.2 | 267.6±78.8 | 146.0±58.3 | 303.6±110.9 | 53.2±19.5 | 94.2±40.0 | 224.0±104.2 | 165.3±78.2 | **35.8±18.2** |
| | F1 | 62.7±7.1 | 64.8±3.7 | 46.2±10.5 | 66.0±7.9 | 5.7±0.7 | 85.4±5.0 | 75.9±2.9 | 41.1±0.8 | 40.4±4.9 | **90.0±3.9** |
| 1 | SHD | 18.2±2.9 | 18.8±5.5 | 40.0±1.6 | 34.0±11.8 | 782.8±35.2 | 10.2±4.9 | 21.2±7.2 | 94.0±11.4 | 120.3±6.6 | **6.2±2.8** |
| | SID | 131.2±64.7 | 139.2±59.1 | 258.3±98.3 | 214.4±118.5 | 344.4±165.1 | 69.8±42.9 | 102.8±67.9 | 152.6±44.7 | 198.6±99.1 | **36.8±21.8** |
| | F1 | 77.4±5.6 | 72.3±7.0 | 47.2±3.5 | 58.1±9.9 | 5.6±0.9 | 86.6±5.7 | 74.2±6.4 | 40.0±1.3 | 34.6±1.7 | **92.3±3.5** |
| Training time | | 29.6±6.5 | 40.1±0.4 | 3.1k±0.2k | 23k±1.8k | 322.6±18.7 | 1.3k±38.4 | 1.3k±59.9 | 71.5±2.6 | 89.1±8.9 | 319.8±98.8 |
| | | | | | **SynER1 (n=1000)** | | | | | | |
| 0 | SHD | 6.6±1.3 | 6.2±2.2 | 3.0±1.4 | 0.8±1.1 | 39.0±1.2 | 0.6±1.2 | 2.6±1.8 | 12.6±3.3 | 11.0±1.6 | **0.4±0.8** |
| | SID | 21.4±12.5 | 24.4±14.4 | 7.6±6.2 | 0.6±1.2 | 14.6±5.6 | 0.6±1.2 | 10.8±11.9 | 8.6±3.6 | 9.2±4.2 | **0.6±1.2** |
| | F1 | 38.4±10.9 | 36.3±24.2 | 78.8±9.5 | 95.9±6.1 | 22.2±4.2 | 96.8±6.3 | 79.6±13.4 | 48.1±6.3 | 52.7±7.4 | **97.8±4.4** |
| 0.25 | SHD | 6.6±1.6 | 6.4±2.1 | 3.3±1.8 | 2.6±2.2 | 39.4±2.5 | 1.6±1.3 | 2.6±1.8 | 9.3±2.6 | 9.2±2.4 | **1.0±1.2** |
| | SID | 20.2±12.4 | 23.0±15.0 | 11.3±3.0 | 7.2±7.7 | 13.4±6.3 | 4.2±3.9 | 12.2±12.1 | 9.3±4.7 | 12.0±6.0 | **2.2±2.8** |
| | F1 | 36.1±22.3 | 36.0±22.7 | 68.8±6.3 | 79.0±15.4 | 20.6±9.2 | 86.6±12.4 | 78.8±14.9 | 50.7±10.2 | 52.5±5.1 | **93.3±8.2** |
| 0.5 | SHD | 6.2±1.9 | 5.0±1.8 | 4.3±1.2 | 3.8±2.7 | 39.4±1.3 | 2.8±1.6 | 3.8±0.9 | 11.3±1.6 | 12.2±1.6 | **2.0±0.8** |
| | SID | 17.0±11.9 | 15.8±8.8 | 13.6±2.0 | 16.6±14.2 | 14.6±10.6 | 8.2±8.9 | 12.2±4.6 | 11.6±3.0 | 11.2±8.1 | **3.2±1.9** |
| | F1 | 44.2±18.8 | 54.1±17.2 | 56.5±10.0 | 67.3±22.4 | 20.7±15.7 | 78.6±9.7 | 64.5±10.8 | 42.8±3.9 | 44.8±6.7 | **86.8±5.1** |
| 0.75 | SHD | 4.2±1.7 | 5.2±2.9 | 5.3±2.0 | 1.8±0.4 | 39.6±2.2 | 2.2±1.1 | 3.6±2.1 | 9.0±2.4 | 11.0±1.6 | **1.4±0.8** |
| | SID | 10.8±8.2 | 18.0±16.1 | 16.3±1.6 | 5.8±4.0 | 17.6±10.9 | 7.6±6.6 | 10.0±6.9 | **4.3±4.1** | 6.0±2.1 | 6.2±6.2 |
| | F1 | 68.9±12.6 | 56.1±25.6 | 49.5±12.1 | 83.5±4.3 | 19.9±4.9 | 81.1±11.1 | 70.3±12.5 | 60.5±6.9 | 50.1±7.6 | **89.3±6.9** |
| 1 | SHD | 3.0±1.0 | 4.4±2.6 | 7.3±4.7 | 2.2±1.4 | 40.0±2.1 | 1.8±0.9 | 3.2±1.9 | 10.6±3.6 | 13.3±2.0 | **1.2±0.7** |
| | SID | 8.4±8.9 | 17.0±16.5 | 16.6±6.0 | 9.6±7.9 | 18.0±10.4 | 9.0±6.1 | 10.6±6.5 | **5.0±4.5** | 6.0±2.1 | 5.6±6.6 |
| | F1 | 79.6±6.7 | 61.5±23.3 | 43.5±17.6 | 77.2±15.6 | 18.5±7.8 | 79.9±11.0 | 71.2±10.5 | 54.5±11.1 | 44.7±8.9 | **90.2±6.7** |
| Training time | | 2.1±0.7 | 32.2±1.3 | 471.5±33.7 | 9.9±0.4 | 151.1±33.9 | 4.0±0.9 | 33.3±2.0 | 36.5±42.6 | 34.9±3.7 | 3.5±0.7 |
| | | | | | **SynER1 (n=5000)** | | | | | | |
| 0 | SHD | 6.4±1.0 | 5.6±1.2 | 0.3±0.4 | **0.0±0.0** | 37.8±1.5 | 0.6±0.4 | 2.8±1.8 | 3.6±1.2 | 3.0±0.0 | 0.4±0.8 |
| | SID | 19.4±9.2 | 18.0±9.8 | 1.3±1.8 | **0.0±0.0** | 8.4±7.8 | 2.4±1.9 | 10.8±8.7 | 6.33±3.8 | 4.3±4.7 | 0.6±1.2 |
| | F1 | 41.1±10.7 | 50.6±5.7 | 97.7±3.1 | **100±0.0** | 26.6±5.5 | 93.2±5.5 | 74.1±13.5 | 71.8±13.3 | 80.6±2.6 | 96.8±6.3 |
| 0.25 | SHD | 6.2±2.3 | 5.2±1.6 | 1.0±0.8 | 0.6±0.8 | 39.0±1.0 | 2.0±0.6 | 3.0±2.5 | 3.0±1.6 | 3.0±2.1 | **0.0±0.0** |
| | SID | 17.0±10.7 | 16.4±11.2 | 3.6±2.6 | 2.2±2.7 | 11.8±7.1 | 6.0±2.8 | 11.8±9.7 | 8.0±4.3 | 6.0±2.1 | **0.0±0.0** |
| | F1 | 39.9±28.1 | 57.3±12.5 | 91.0±6.3 | 94.6±6.5 | 22.2±3.8 | 81.1±6.7 | 75.1±19.5 | 70.2±19.9 | 81.0±10.7 | **100±0.0** |
| 0.5 | SHD | 5.4±1.8 | 5.4±2.2 | 2.0±0.8 | 1.6±1.0 | 39.4±0.4 | 4.6±1.6 | 4.4±3.6 | 2.3±1.2 | 4.3±3.2 | **1.4±1.0** |
| | SID | 12.8±7.3 | 16.4±11.4 | 6.3±1.2 | **2.2±2.7** | 16.6±8.0 | 14.0±6.4 | 12.4±11.1 | 5.3±1.6 | 6.6±1.2 | 5.4±5.7 |
| | F1 | 54.0±17.2 | 54.0±17.6 | 81.5±7.1 | 84.1±9.9 | 20.8±4.7 | 62.0±14.9 | 68.1±23.9 | 78.3±8.2 | 73.1±10.8 | **87.7±7.6** |
| 0.75 | SHD | 4.2±1.1 | 4.8±3.0 | 4.6±1.8 | 6.2±2.9 | 39.8±1.7 | 5.4±2.6 | 6.0±2.6 | 5.6±3.3 | 6.0±2.8 | **2.2±2.0** |
| | SID | 11.2±7.6 | 17.2±16.1 | 15.0±2.1 | 20.8±9.5 | 19.4±11.1 | 14.2±6.9 | 17.2±12.1 | 8.0±4.9 | 8.0±7.0 | **4.4±2.8** |
| | F1 | 69.0±6.7 | 56.8±26.9 | 54.8±17.9 | 48.6±25.2 | 19.2±6.0 | 62.5±14.5 | 56.4±22.8 | 65.4±14.5 | 64.3±11.6 | **84.0±10.8** |
| 1 | SHD | **3.2±0.9** | 4.8±2.4 | 9.0±3.5 | 6.2±1.6 | 40.4±1.3 | 6.6±3.7 | 6.2±2.2 | 5.3±2.8 | 5.6±3.6 | 3.4±1.9 |
| | SID | 9.0±8.6 | 18.4±14.9 | 20.3±5.1 | 17.0±8.7 | 22.0±8.2 | 17.0±6.9 | 17.8±9.6 | **4.3±1.8** | 4.6±0.9 | 10.2±3.5 |
| | F1 | **77.5±6.8** | 56.1±20.4 | 35.3±3.7 | 48.5±19.3 | 17.0±4.7 | 52.3±14.0 | 48.6±21.5 | 69.3±8.8 | 69.2±14.7 | 74.7±10.6 |
| Training time | | 3.9±0.6 | 32.7±0.6 | 551.1±90.5 | 55.7±1.1 | 458.5±194.5 | 14.3±0.8 | 42.4±13.5 | 65.5±33.3 | 36.6±2.7 | 15.0±0.4 |

model. Here, we explore the speedup of pre-pruning on different datasets and sparsities. Fig. 8 shows that our method effectively speeds up the pruning time on all datasets through pre-pruning. The symbols *-L* and *-N* in the dataset denote linear and nonlinear, respectively. Comparing SynER1 and SynER4, we can find that acceleration is more effective in sparse causal graphs than in dense ones. This is due to the fact that in sparse graphs, there are more edges with low confidence in parent score, and pre-pruning is more efficient. Moreover, the acceleration ratio of pre-pruning becomes more significant when the number of nodes $d$ grows, which makes a major contribution to the speed advantage of CaPS in larger-scale datasets.

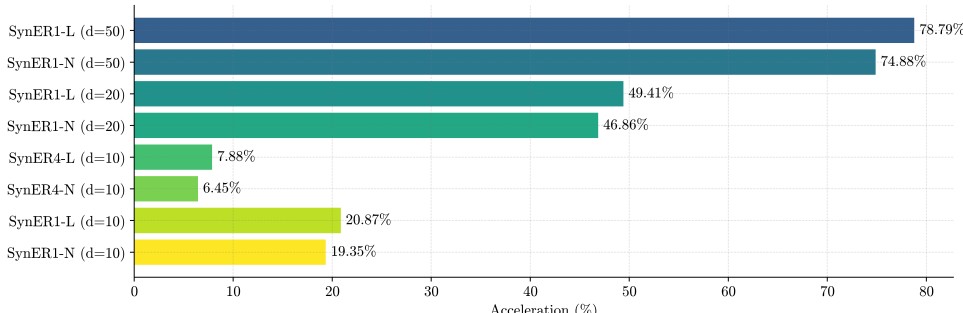

Figure 8: Percentage of acceleration using pre-pruning.

## C.6    Order Divergence

**Order divergence.** Order divergence [11] is a measure of the discrepancy between the estimated topological ordering $\pi$ and the adjacency matrix of the true causal graph $\mathcal{A}$, which is expressed as:

$$D_{top}(\pi, \mathcal{A}) = \sum_{i=1}^{d} \sum_{j:\pi_i > \pi_j} \mathcal{A}_{i,j} \qquad (25)$$

Here, we compare two-stage ordering-based methods in datasets with different linear proportions and sparsity. The results in Figure 9 show that CaPS has a much lower order divergence than *sortnregress* in different linear proportions and sparsity, suggesting that variance is not a reliable measure of topological ordering in our synthetic datasets. CaPS consistently achieves the best order divergence in most of settings, which reflects the effectiveness of our method in recognizing the correct permutation.

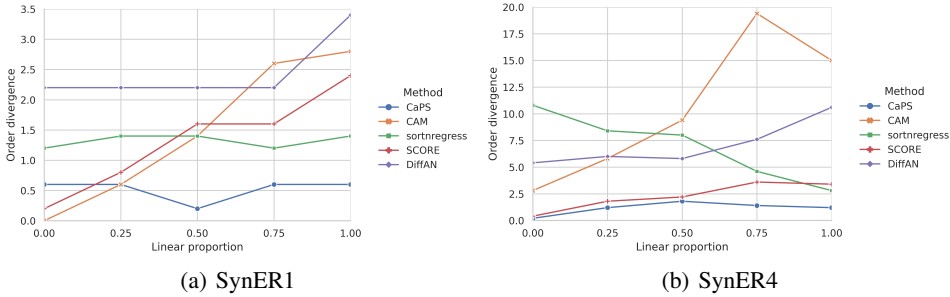

(a) SynER1                                        (b) SynER4

Figure 9: Order Divergence of SynER1 and SynER4 with different linear proportions and sparsity.

## C.7    Beyond Our Assumption

Theoretically, according to the sufficient condition $(ii)$ of Assumption 1, our methods also has a good potential for unequal variance settings. We experimentally find that CaPS performs well under both Gaussian noise with unequal variance and non-Gaussian noise, as illustrated in Figs. 10 and 11. More relaxed conditions for CaPS can be explored in future work.

**Unequal variances noise.** Beyond the equal variance noise, we generated datasets with unequal variance Gaussian noise. Specifically, following the most popular settings in previous works, for each variable $x_i$, the corresponding $\epsilon_i \sim \mathcal{N}(0, \sigma_i^2)$, where $\sigma_i^2$ are independently sampled uniformly in $U(0.4, 0.8)$. Fig. 10 shows the experiment results of eight baselines. We can observe that CaPS performs second only to CAM for sparser (SynER1) graphs. However, CAM does not handle dense graphs well. For denser (SynER4) graphs, our method performs similarly to the best methods SCORE in almost all ranges.

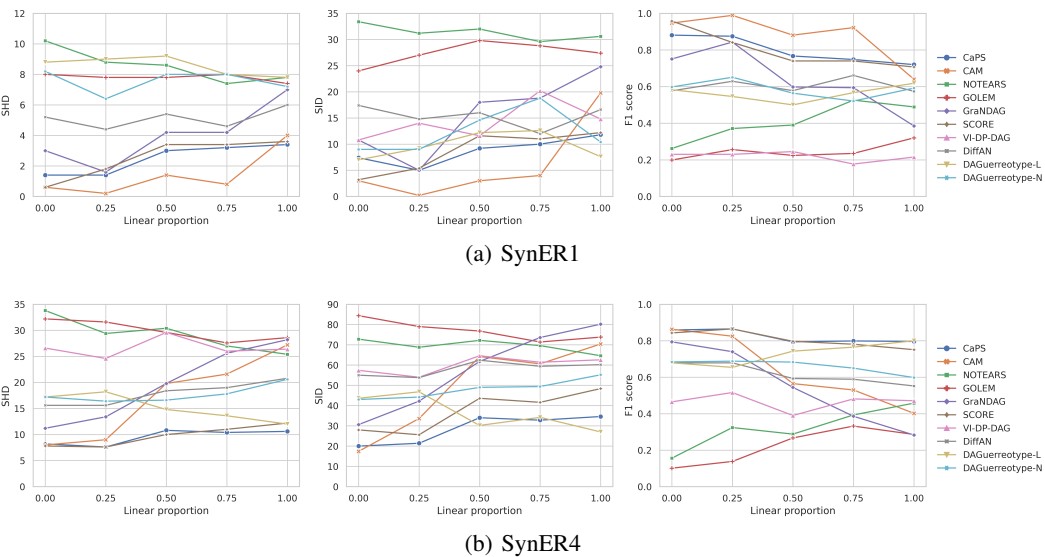

(a) SynER1

(b) SynER4

Figure 10: Results of SynER1 and SynER4 with unequal variance Gaussian noise.

**Non-Gaussian noise.** We also experimentally explore the potential of CaPS for ANM with non-Gaussian noise. Specifically, for each variable $x_i$, we set the noise distribution to be Gumbel and Laplace. Fig. 11 shows the experiment results of eight baselines. We can observe that CaPS performs consistently well for both Gumbel and Laplace noise in almost all ranges, especially when the linear proportions are greater than 0.25.

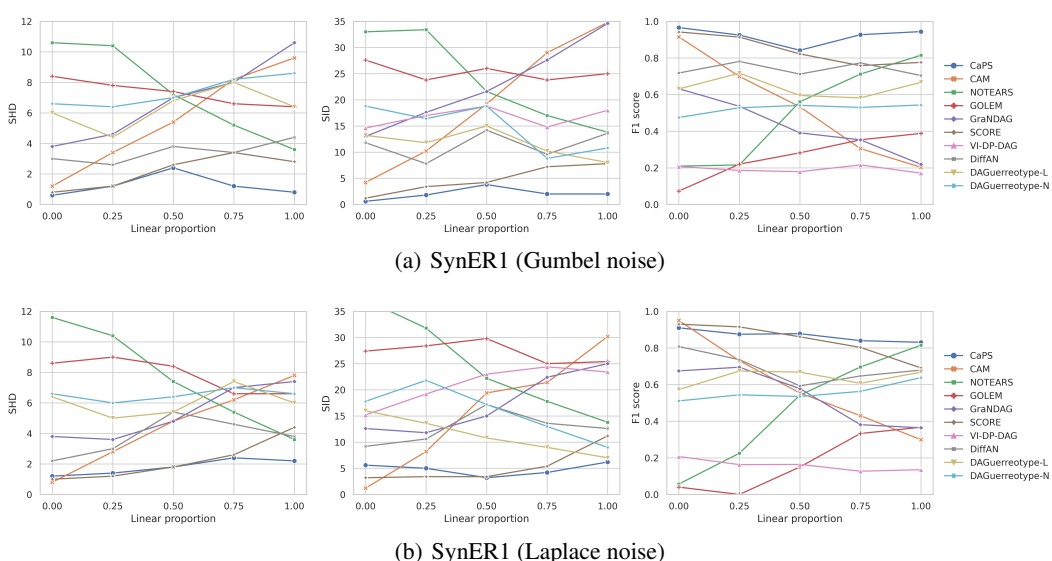

(a) SynER1 (Gumbel noise)

(b) SynER1 (Laplace noise)

Figure 11: Results of SynER1 with different non-Gaussian noise.

