# OpenReview forum: "Ordering-Based Causal Discovery for Linear and Nonlinear Relations"
_NeurIPS.cc/2024/Conference — NeurIPS 2024 poster_

### Official Review · Reviewer_7cyK · 2024-07-06

**Soundness:** 3
**Presentation:** 3
**Contribution:** 2
**Rating:** 6
**Confidence:** 3

**Summary:**

This paper studies the causal discovery problem in mixed functional relations data, where both linear and non-linear relationships exist in the causal graph. The author presents a Jacobian score-based method (essentially a score-matching method) to identify leaf nodes and thereby recover the causal order. The experimental results demonstrate the efficiency of the proposed methods.

**Strengths:**

1. The paper is clearly written and well-organized.

2. The setting of mixed functional relations data is interesting and may be important for real-world scenarios.

3. The author proposes a Jacobian score-based method, which is an extension of the score-matching method for non-linear Additive Noise Models (ANM).

**Weaknesses:**

1. The non-decreasing variance of noises assumption is too strong and restrictive. Typically, in ANM, the noise term is assumed to be mutually independent.

2. It appears that the primary difference between the score-matching method for ANM and the proposed method is the introduction of Assumption 1.

3. If I use an independent residuals-based method, it seems to work in your setting. So, what are the advantages of the proposed method? For example, is the proposed method capable of handling large-scale structures? If so, the experimental results should demonstrate this.

**Questions:**

See Weaknesses.

**Limitations:**

NAN

---

> ### Author Rebuttal · Authors · 2024-08-06
>
> Your insightful advice is greatly valued, as it can contribute to the progress of our work. We hope that the answers provided below have resolved your inquiries.
>
> **Q4.1:** non-decreasing variance too strong.
>
> **A4.1:** Before answering this question about assumptions, we want to highlight some facts of ANM. For an ANM $y=f(x)+\epsilon$, we have to make some assumptions on $f$ or $\epsilon$ due to the problem of backward model (see Prop.23 in ref. [9]). This paper does not make any additional assumptions on $f$ (linear or nonlinear), therefore, some assumptions on $\epsilon$ are inevitable. To the best of our knowledge, our assumptions is the weakest assumption that works well under both linear and nonlinear ANM.
>
> Then, it is necessary to emphasise that CaPS works under (i) **or** (ii) in assumption 1. So, our assumption is **weaker** than non-decreasing variance. CaPS is able to work well using the condition (ii) if non-decreasing variance does not hold. For example, considering a variance-unsortable scenario with $\sigma^2\sim U(1,10)$ and causal effect greater than 0.9, CaPS can also work well because the the sum of parent score is greater than the given lower bound in condition (ii). In other words, non-decreasing variance is just one of the scenarios in which CaPS works well, and CaPS can also handle many variance-unsortable scenarios.
>
> **Q4.2:** The primary difference between the SCORE and CaPS is assumption 1.
>
> **A4.2:** We understand your concerns, but SCORE and CaPS is totally different for the following reasons:
>
> (1) More generalized scenarios. CaPS aims to handle both linear and nonlinear and most possibly mixed relations while SCORE only handles nonlinear relations. CaPS is the first ordering-based method to deal with this scenario.
>
> (2) Different theoretical ideas. As mentioned in A4.1, SCORE is derived through the properties of nonlinear $f$ while CaPS derives Theorem 1 by the properties of $\epsilon$, which is a totally different ideas in theoretical perspective (see details in App. A.2).
>
> (3) New concept for identifiability and post-processing. This paper proposes a new concept of "parent score", which gives a new lower bound for identifiability with causal effect (see Corollary 1). This concept can be used to accelerate the pruning process (see Fig. 8) and correct inaccurate predictions in the pruning step (see Fig. 4 and 7).
>
> **Q4.3:** independent residuals-based method work in this setting? what are the advantages of CaPS? (e.g. handing large-scale structures)
>
> **A4.3:** Thanks for the kindly reminder of comparision of the conditional independence (CI) test-based methods. About the independent residuals-based method, we're not sure exactly which method you mean, but we guess it's ReCIT[1, 2]. Since the source code of ReCIT is not available for comparision, we adopt the closest CI-based baseline KCIT[3] compared with ReCIT.
>
> Yes, as you mentioned, although these two CI test-based methods not explicitly need on linear and non-linear assumptions, they have high computional complexity. They need to search the solution on a fully connect DAG while ordering-based method only need a smaller search space with the topological ordering. This make CI test-based methods ususally has exponential complexity and hard to implement in a large-scale structures. In Table 5, we show the performance of CaPS vs KCIT and there are two conclusions we can learn from these experimental results.
>
> (1) Although KCIT does not significantly degrade in linear and nonlinear performance, it consistently underperforms CaPS in all scenarios with different linear rate.
>
> (2) For capablity of handling large-scale structures, CaPS is consistently better than KCIT with significantly outperforming metrics and faster training speeds (e.g. 18 seconds vs 2 hours in Table 5). In addition, even compared to the ordering-based approach, CaPS are able to better handle the large-scale structures (see Fig.3, App. C.5 and Fig. 8).
>
> *Since KCIT and ReCIT are compatible with CaPS, KCIT and ReCIT can use CaPS to largely reduce their search space.
>
> This comparison further shows the advantages of CaPS and we will add these discussion of ReCIT,KCIT and CaPS to our related work and experiments in the latest version.
>
> [1] Zhang H et al. (2018) Measuring conditional independence by independent residuals: theoretical results and application in causal discovery.
>
> [2] Zhang H et al. (2019) Measuring conditional independence by independent residuals for causal discovery.
>
> [3] Zhang K et al. (2011) Kernel-based Conditional Independence Test and Application in Causal Discovery.

---

> > ### Comment · Reviewer_7cyK · 2024-08-11
> > **Thank you for your response**
> >
> > Regarding the independent residuals-based method, I am referring to the traditional nonlinear Additive Noise Model (ANM), where the causal direction is identified by testing the independence between the residual and its parent variables.
> >
> > Additionally, I am satisfied with the responses to the other questions, so I am raising my score accordingly.
> >
> > By the way, it seems that during the author-rebuttal stage, it is not allowed to use the 'official comment' button. For example, the author can use the general rebuttal button to submit supplementary experiments. I overlooked this issue here.

---

> > > ### Author Response · Authors · 2024-08-11
> > > **Thanks for your feedback**
> > >
> > > Thank you for revising your score and for your thoughtful review. Sorry for the inappropriate use of the 'official comment', and thank you again for overlooking this issue.  We greatly appreciate your recognition of our efforts.
> > >
> > > About independent residuals-based method, we get your point with your additional interpretations. This method will often be used to identify the direction of causal sub-structures, e.g. chain, fork and collider, which is similar to CI test-based method. As you already pointed out, although this method may not significantly degrade in linear and nonlinear performance, it is not capable of handling large-scale structures. As this paper [4] states and our additional experiments in Table 4, the daunting cost of checking every candidate sub-structure is intolerable. Thus, CaPS will consistently have better capablity of handling large-scale structures compared with this method. And, as stated in the rebuttal above, the complexity of this method can be greatly reduced by CaPS since this method are compatible with CaPS.
> > >
> > > [4] He Y et al. (2021) DARING: Differentiable Causal Discovery with Residual Independence

---

> ### Author Response · Authors · 2024-08-06
> **Table 4: comparison of CaPS and CI-based methods**
>
> | dataset                 | Linear rate | KCIT            |            |             | CaPS       |            |              |
> |-------------------------|-------------|-----------------|------------|-------------|------------|------------|--------------|
> |                         |             | SHD             | SID        | F1          | SHD        | SID        | F1           |
> | SynER1 d=10             | 0           | 4.2±0.4         | 18.8±9.8   | 0.617±0.050 | 0.6±0.8    | 4.2±7.9    | 0.958±0.061  |
> |                         | 0.25        | 4.8±0.4         | 19.6±8.1   | 0.541±0.126 | 0.8±0.7    | 5.6±7.6    | 0.944±0.057  |
> |                         | 0.5         | 4.8±0.4         | 16.2±7.9   | 0.573±0.070 | 0.6±0.8    | 1.6±2.7    | 0.961±0.055  |
> |                         | 0.75        | 5.4±2.1         | 20.0±13.0  | 0.546±0.166 | 0.8±1.1    | 3.0±4.2    | 0.924±0.098  |
> |                         | 1           | 4.8±1.9         | 17.6±12.2  | 0.588±0.146 | 1.2±1.1    | 3.6±4.0    | 0.901±0.090  |
> | Training time (seconds) |||| 308.2±191.3 ||| 8.02±1.08 |
> | SynER1 d=20             | 0           | 13.0±4.14       | 102.6±49.9 | 0.444±0.171 | 0.8±0.40   | 3.4±2.87   | 0.981±0.010  |
> |                         | 0.25        | 11.4±2.3        | 91.0±38.6  | 0.528±0.137 | 1.2±1.17   | 7.6±10.25  | 0.960±0.039  |
> |                         | 0.5         | 10.8±2.5        | 75.4±20.7  | 0.596±0.103 | 4.2±3.05   | 17.0±11.48 | 0.949±0.046  |
> |                         | 0.75        | 12.4±2.1        | 81.4±15.2  | 0.538±0.112 | 1.6±1.62   | 11.8±10.91 | 0.949±0.046  |
> |                         | 1           | 11.0±2.5        | 69.2±16.3  | 0.593±0.137 | 2.0±2.28   | 12.4±12.14 | 0.937±0.060  |
> | Training time (seconds) |||| 6551.6±1570.6 ||| 15.84±3.36 |
> | SynER1 (d=50)           | 0    | 40 | 549 | 0.37  | 7.2±4.44     | 56.6±49.01 | 0.914±0.062  |
> |                         | 0.25 | 39 | 549 | 0.377 | 11.4±1.85    | 68.4±18.63 | 0.857±0.033  |
> |                         | 0.5  | 40 | 550 | 0.367 | 11.4±4.49    | 72.8±47.52 | 0.865±0.054  |
> |                         | 0.75 | 26 | 325 | 0.62  | 8.2±3.06     | 35.8±18.92 | 0.900±0.039  |
> |                         | 1    | 40 | 422 | 0.444 | 6.2±2.85     | 36.8±21.87 | 0.923±0.035  |
> | Training time (seconds) ||||$\geq 12h$| ||319.85±98.82|
>
> Due to the long training time of KCIT at 50 nodes, we only report its performance with one trial in Table 4. Other results are reported with 5 trials.

---

### Official Review · Reviewer_SPoa · 2024-07-09

**Soundness:** 3
**Presentation:** 3
**Contribution:** 3
**Rating:** 6
**Confidence:** 3

**Summary:**

This paper proposes an ordering based causal discovery method when the underlying causal model has both linear and nonlinear causal relationships. Starting with a method to iteratively find leaf nodes, this paper proposes to use parent score for better pruning. Results show that the proposed method outperforms baselines.

**Strengths:**

1. Paper is written well and easy to understand.
2. Theoretical motivations are clearly explained and the proof are adequately provided.
3. Experiments are extensive and cover all theoretical aspects.

**Weaknesses:**

1. Results are not great on real-world datasets.
2. Topological divergence is a popular metric for evaluating the topological order. Very few results are presented in supplementary on this metric.

**Questions:**

Please see weaknesses section.

**Limitations:**

Limitations are discussed.

---

> ### Author Rebuttal · Authors · 2024-08-06
>
> We appreciate your constructive feedback, as it can help us improve our work. We trust that the following answers have clarified your questions.
>
> **Q3.1:** Results are not great on real-world datasets.
>
> **A3.1:** Yes, CaPS achieves the best performance on Sachs dataset but only the second best in Syntren. We have analyzed the source of the discrepancy in Appendix C.1, the pattern of Syntren is very special containing many star networks with short topological dependency path. This pattern is not friendly to ordering-based methods (see details in Fig. 5 and lines 524-528). Despite this unfavourable dataset, CaPS achieves the second in all baseline and the best performance in all ordering-based methods. We believe, in a sense, this fact reinforces the effectiveness of CaPS.
>
> **Q3.2:** Few results of topological divergence
>
> **A3.2:** We already shows the order divergence of SynER1 and SynER4 in App. C.6. To further address your concerns, we show more results of this metric in different datasets in Table 4. The conclusion is CaPS consistently achieves the best order divergence under the datasets with different DAG tpye, sparsity, scale and sample sizes. We will update it to the latest version of our manuscript.

---

> ### Author Response · Authors · 2024-08-06
> **Table 4: more results of order divergence**
>
> **Table 4. More results of order divergence.**
> | Dataset        | linear_rate | sortnregress | SCORE       | DiffAN  | CAM         | CaPS        |
> | -------------- | ----------- | ------------ | ----------- | ------- | ----------- | ----------- |
> | SynSF1         | 0           | 1.4±1.4      | **0.0±0.0** | 3.2±2.7 | 0.2±0.4     | 0.2±0.4     |
> |                | 0.25        | 1.6±1.3      | **0.2±0.4** | 3.0±2.6 | 1.4±0.4     | **0.2±0.4** |
> |                | 0.5         | 1.6±1.0      | 1.2±1.1     | 3.0±2.0 | 1.2±0.7     | **0.0±0.0** |
> |                | 0.75        | 2.0±1.6      | 1.4±1.3     | 2.6±1.6 | 2.4±1.8     | **0.2±0.4** |
> |                | 1           | 2.2±1.3      | 2.2±2.2     | 2.6±1.4 | 4.4±1.9     | **0.2±0.4** |
> | SynSF4         | 0           | 6.6±1.6      | 0.6±1.2     | 4.4±2.6 | 2.0±1.6     | **0.4±0.8** |
> |                | 0.25        | 4.8±1.9      | **0.6±1.2** | 4.4±1.8 | 3.8±4.0     | 1.0±1.2     |
> |                | 0.5         | 6.2±1.9      | 2.2±1.4     | 4.6±1.6 | 6.0±2.8     | **1.4±1.0** |
> |                | 0.75        | 4.2±1.7      | 2.8±1.4     | 4.6±1.7 | 9.6±2.8     | **1.8±1.1** |
> |                | 1           | 3.2±0.9      | 2.8±1.1     | 7.6±2.5 | 13.0±3.5    | **1.2±1.1** |
> | SynER1(d=20)   | 0           | 3.6±2.0      | 0.2±0.4     | 2.4±1.4 | 0.6±0.4     | **0.0±0.0** |
> |                | 0.25        | 3.2±1.4      | **0.2±0.4** | 3.2±1.1 | 2.4±1.3     | 0.6±0.4     |
> |                | 0.5         | 3.2±0.7      | 1.2±1.6     | 5.0±1.4 | 2.6±0.8     | **1.2±1.1** |
> |                | 0.75        | 2.6±0.4      | 1.2±0.7     | 3.4±1.3 | 6.4±1.4     | **0.8±0.7** |
> |                | 1           | 2.4±1.0      | 2.0±0.6     | 3.8±1.1 | 7.8±1.9     | **1.0±0.8** |
> | SynER1(n=1000) | 0           | 1.6±0.4      | **0.0±0.0** | 1.8±1.1 | **0.0±0.0** | **0.0±0.0** |
> |                | 0.25        | 1.0±0.6      | 0.2±0.4     | 2.8±1.1 | 1.0±0.6     | **0.0±0.0** |
> |                | 0.5         | 1.0±1.0      | 1.4±1.0     | 2.0±0.8 | 2.2±1.7     | **0.4±0.4** |
> |                | 0.75        | 1.4±1.4      | 1.0±0.8     | 2.8±1.9 | 1.2±0.7     | **0.6±0.8** |
> |                | 1           | 1.4±1.4      | 1.6±0.8     | 2.6±1.0 | 1.8±1.1     | **0.6±0.7** |

---

> > ### Comment · Reviewer_SPoa · 2024-08-12
> >
> > I thank the authors for their response. I've read their response and I will stay with my score.

---

> > > ### Author Response · Authors · 2024-08-12
> > > **Thanks for your feedback**
> > >
> > > Thank you very much for your valuable suggestions and insightful comments on our manuscript. We would like to express our sincere gratitude for your recognition of our efforts . If you have any further questions, please feel free to reach out.

---

### Official Review · Reviewer_8ao9 · 2024-07-20

**Soundness:** 4
**Presentation:** 3
**Contribution:** 3
**Rating:** 6
**Confidence:** 4

**Summary:**

The authors propose an ordering-based causal discovery algorithm designed to handle both linear and nonlinear causal relations in an SEM. In contrast to existing methods that assume purely linear or nonlinear relations, CaPS introduces a unified criterion for topological ordering and a new "parent score" to quantify the average causal effect, which aids in pruning and correcting predictions. Experimental results show that CaPS outperforms some sota methods on synthetic data with mixed linear and nonlinear relations and demonstrates competitive performance on real-world data.

**Strengths:**

* CaPS provides a new approach that can handle both linear and nonlinear causal relationships, addressing a relevant gap in current causal discovery methods.
* The introduction of the parent score is interesting and provides a quantitative measure of causal strength, which improves the pruning process and prediction accuracy.
* The authors present a new criterion for distinguishing leaf nodes using the expectation of the Hessian of the data log-likelihood and provides sufficient conditions for the identifiability of the causal graph, inspired by SCORE and LiSTEN.

**Weaknesses:**

* All noises are assumed to be Gaussian.
* The identifiability conditions rely on assumptions such as non-decreasing variance of noises, which is hard to hold in practical scenarios.
* Some more recent methods are not compared against.

**Questions:**

* Looking at the derivations, the approach seems difficult to generalize to more general noises. What are your thoughts on this?
* Both conditions in Theorem 1 seem impossible to verify, is this sentiment correct?
* Did you assume equal variances in the experiments? I think experimenting on settings where noise variances are random might make sense in this case.
* I think a couple of more recent methods such as DAGMA (Bello et al. 2022) and TOPO (Deng et al. 2023) are known to outperform both NOTEARS and GOLEM. I think it could be worth comparing against those methods.

Bello et al. (2022), "DAGMA: Learning DAGs via M-matrices and a Log-Determinant Acyclicity Characterization".

Deng et al. (2023), "Optimizing NOTEARS objectives via topological swaps"


* Line 52: "creterion" should be "criterion."

**Limitations:**

There are some limitations not "explicitly" stated such as assumptions on causal sufficiency and Gaussianity of noises.

---

> ### Author Rebuttal · Authors · 2024-08-06
>
> Thank you for your valuable suggestions, as they can aid in enhancing our work. We hope that the responses below have addressed your concerns.
>
> **Q2.1:** non-decreasing variance of noise
>
> **A2.1:** The first thing we need to emphasise is that CaPS works under (i) **or** (ii) in assumption 1. So, our assumption is **weaker** than non-decreasing variance. CaPS is able to work well using the condition (ii) if non-decreasing variance does not hold. For example, considering a variance-unsortable scenario with $\sigma^2\sim U(1,10)$ and causal effect greater than 0.9, CaPS can also work well because the the sum of parent score is greater than the given lower bound in condition (ii). In other words, non-decreasing variance is just one of the scenarios in which CaPS works well, and CaPS can also handle many variance-unsortable scenarios.
>
> **Q2.2:** Gaussian assumption & derivations seems difficult to generalize to more general noise
>
> **A2.2:** Before answering the this question about assumptions, we want to highlight some facts of ANM. Without interventional data, all ANM-based models ($y=f(x)+\epsilon$) have to make some assumptions on $f$ or $\epsilon$ due to the problem of backward model (see Prop.23 in ref. [9]). Some paper make purely linear or nonlinear assumption on $f$, e.g., SCORE and LISTEN. This paper does not make any additional assumptions on $f$ (linear or nonlinear), therefore, some assumptions on $\epsilon$ are inevitable. To the best of our knowledge, our assumptions is the weakest assumption that works well under both linear and nonlinear ANM.
>
> Theoretically, the Gaussian assumption is necessary, which are used in Eq. 10 to prove Theorem 1. However, experimentally, the results in App. C.7 shows that CaPS performs consistently well under non-Gaussian noise (Gumbel and Laplace), which shows the potential of CaPS generalization to other noises. We are still working to find a more elegant solution to relax the $\epsilon$ to non-Gaussian noise. However, we believe that CaPS, as the first ordering-based method capable of handling both linear and nonlinear, contributes sufficiently to the field.
>
> **Q2.3:** Both conditions in Theorem 1 seem impossible to verify.
>
> **A2.3:** We understand your concerns and this sentiment is right, but most of conditions is unverifiable without the ground truth SEM. In synthetic data, we can easily verify our conditions since we have the real parameters of $f$ and $\epsilon$. In the real-world application, as shown in Fig. 1, we can't even verify $f$ is linear and non-linear since we do not have the ground truth SEM.
>
> Thus, we need a method with the conditions which more likely to be close to the real-world patterns while many conditions is usually unverifiable. CaPS is able to solve this problem better than existing works based on the following reason:
>
> (1) CaPS can work well in both linear and nonlinear and most possibly mixed cases, which is widespread in real-world data.
>
> (2) CaPS gives a new lower bound for identifiable causal effects in Assumption 1(ii), which can be use to discovery non-weak causal relations in real-world data.
>
> (3) Experimentally, CaPS is an outperforming ordering-based method under different tpyes of synthetic datasets and real-world datasets. Especially, our experiment results show CaPS can effectively support non-Gaussian noise (Gumbel and Laplace) in addtion to theoretically proved Gaussian noise.
>
> **Q2.4:** assume equal variances in the experiments?
>
> **A2.4:** We consider the scenario both equal variances and unequal variances in our manuscript. The results of unequal variances are given in App. C.7. Following the most popular settings in previous works, for each variable $x_i$, the noise $\epsilon_i\sim N(0, \sigma_i^2)$, where $\sigma_i^2$ are independently sampled uniformly in $U(0.4, 0.8)$ . Under this random variances scenario, CaPS still achieved the best or competitive performance under unequal variances noise.
>
> **Q2.5:** comparing with DAGMA and TOPO.
>
> **A2.5:** Thank you for sharing these two strong continuous optimization baselines. With their wonderful open source code, we successfully implemented them in our experimental scenarios. Since this paper considers both linear, non-linear and mixed scenarios, for a fair comparison, we compare the performance of CaPS and their linear (-L) and nonlinear (-N) versions, respectively, in Table 3. All parameters and settings of these two baseline follow their original manuscript or source code.
>
> Yes, as you mentioned, DAGMA and TOPO are stronger baselines than NOTEARS and GOLEM. And, we can learn two conclusions from Table 3:
>
> (1) Compared with DAGMA and TOPO, CaPS can consistently achieves best performance under both synthetic and real-world data with different sparsity and linear rate.
>
> (2) Similar to other baselines, DAGMA and TOPO are also suffering significant performance loss when linear/nonlinear assumptions mismatch.
>
> This comparison is important but does not affect any conclusions of this paper. We will update it to the latest version of our manuscript in the related work and experiment.

---

> ### Author Response · Authors · 2024-08-06
> **Table 3: addtional baseline DAGMA and TOPO**
>
> **Table 3. Addtional baselines**
> | dataset | Linear rate | Metrics | DAGMA-L     | DAGMA-N     | TOPO-L      | TOPO-N      | CaPS         |
> |---------|-------------|---------|-------------|-------------|-------------|-------------|--------------|
> | SynER1  | 0           | SHD     | 6.0±1.4     | 4.0±1.7     | 6.8±1.4     | 22.6±11.8   | **0.6±0.8**      |
> |         |             | SID     | 16.0±6.4    | 11.4±6.2    | 16.8±7.9    | 17.4±6.2    | **4.2±7.9**      |
> |         |             | F1      | 0.477±0.118 | 0.684±0.183 | 0.446±0.107 | 0.282±0.069 | **0.958±0.061**  |
> |         | 0.25        | SHD     | 5.2±2.0     | 4.6±1.8     | 5.6±1.4     | 23.2±12.1   | **0.8±0.7**      |
> |         |             | SID     | 15.4±7.4    | 16.2±4.9    | 14.6±9.83   | 16.8±8.7    | **5.6±7.6**      |
> |         |             | F1      | 0.580±0.166 | 0.589±0.170 | 0.582±0.120 | 0.312±0.033 | **0.944±0.057**  |
> |         | 0.5         | SHD     | 4.2±2.3     | 3.4±2.2     | 3.6±2.3     | 18.4±14.1   | **0.6±0.8**      |
> |         |             | SID     | 12.0±7.8    | 11.2±7.1    | 10.6±7.9    | 14.6±10.3   | **1.6±2.7**      |
> |         |             | F1      | 0.681±0.194 | 0.708±0.161 | 0.739±0.175 | 0.429±0.161 | **0.961±0.055**  |
> |         | 0.75        | SHD     | 3.2±1.9     | 3.0±2.2     | 3.4±1.8     | 8.8±4.3     | **0.8±1.1**      |
> |         |             | SID     | 8.8±8.9     | 9.0±9.8     | 8.8±8.9     | 16.6±6.8    | **3.0±4.2**      |
> |         |             | F1      | 0.778±0.135 | 0.760±0.155 | 0.768±0.129 | 0.452±0.058 | **0.924±0.098**  |
> |         | 1           | SHD     | 2.4±1.4     | 3.4±2.05    | 2.4±1.4     | 8.0±4.8     | **1.2±1.1**      |
> |         |             | SID     | 7.6±9.4     | 12.4±9.9    | 7.6±9.4     | 16.2±6.5    | **3.6±4.0**      |
> |         |             | F1      | 0.844±0.097 | 0.719±0.172 | 0.844±0.097 | 0.506±0.105 | **0.901±0.090**  |
> | SynER4 | 0    | SHD | 31.6±1.0    | 32.4±1.4    | 31.6±1.3    | 28.6±4.1    | **14.6±2.4**     |
> |        |      | SID | 67.6±3.9    | 75.8±7.1    | 69.8±5.1    | 70.2±10.6   | **26.2±2.7**     |
> |        |      | F1  | 0.138±0.050 | 0.087±0.024 | 0.129±0.020 | 0.298±0.194 | **0.728±0.040**  |
> |        | 0.25 | SHD | 27.2±4.7    | 25.6±4.8    | 22.8±2.4    | 25.4±4.4    | **12.4±1.9**     |
> |        |      | SID | 60.0±7.7    | 60.6±16.0   | 57.2±6.9    | 65.8±7.4    | **26.8±3.5**     |
> |        |      | F1  | 0.313±0.185 | 0.354±0.214 | 0.476±0.108 | 0.470±0.133 | **0.763±0.046**  |
> |        | 0.5  | SHD | 25.2±2.2    | 22.6±2.3    | 13.0±4.3    | 24.4±2.0    | **10.8±2.6**    |
> |        |      | SID | 61.8±8.9    | 58.6±6.8    | 37.6±9.6    | 72.0±4.6    | **27.2±10.2**    |
> |        |      | F1  | 0.398±0.110 | 0.482±0.098 | 0.755±0.088 | 0.440±0.055 | **0.791±0.059**  |
> |        | 0.75 | SHD | 14.2±4.8    | 17.4±2.8    | 13.0±4.3    | 23.6±2.4    | **6.6±3.5**      |
> |        |      | SID | 46.6±14.0   | 57.0±12.2   | 37.6±9.6    | 67.2±10.5   | **20.0±10.5**    |
> |        |      | F1  | 0.719±0.102 | 0.624±0.078 | 0.755±0.088 | 0.488±0.064 | **0.876±0.069**  |
> |        | 1    | SHD | 9.6±3.8     | 14.0±3.5    | 8.4±2.4     | 21.2±3.0    | **3.2±1.8**      |
> |        |      | SID | 38.4±9.3    | 51.6±13.1   | 33.0±9.0    | 65.6±6.5    | **14.6±9.5**     |
> |        |      | F1  | 0.817±0.089 | 0.709±0.076 | 0.855±0.056 | 0.524±0.099 | **0.936±0.035**  |
> | sachs  | /    | SHD | 13.0±0.0    | 17.0±0.0    | 21.6±0.4    | 47.4±2.3    | **11.0±0.0**     |
> |        |      | SID | 46.0±0.0    | 53.0±0.0    | 44.0±0.0    | **38.6±5.08**   | 42.0±0.0     |
> |        |      | F1  | 0.370±0.0   | 0.0±0.0     | 0.303±0.0   | 0.211±0.064 | **0.5±0.0**     |
>
> Even under purely linear/nonlinear, the experimental results differ from their original manuscripts because of the different settings of synthetic data generation. One different setting is that our nonlinear function is "gp", while they are "mlp". Another different setting is DAG weights, which we set to $[-1, -0.1]\cup[0.1, 1]$ while they are $[-2, -0.5]\cup[0.5, 2]$. In our setup, it would be more difficult to identify DAGs effectively due to the weaker strength of the causal effect.

---

> > ### Comment · Reviewer_8ao9 · 2024-08-13
> >
> > I thank the authors for their response. The additional experiments are helpful. I am not so convinced about A2.2. "To the best of our knowledge, our assumptions is the weakest assumption that works well under both linear and nonlinear ANM." As far as I can tell, LiNGAMs are identifiable, and so are nonlinear models with Gaussian and non-Gaussian noises. Thus, ANM with non-Gaussian noises should also be identifiable. I will keep my score for now.

---

> ### Author Response · Authors · 2024-08-13
> **Thanks for your feedback**
>
> Thank you for your valuable comments and recognition of our efforts, but there seems to be some misunderstanding regarding A2.2.
>
> What we are trying to convey is that the assumption of CaPS is the weakest assumption that **can handle linear, nonlinear and even mixed relations simultaneously**. As you point out, LiNGAM can work under non-Gaussian noises. However, it can **only handle purely linear causal relations**. To further address your concerns, we provide the comparison of CaPS and LiNGAM under SynER1 with Gumbel noise in Table 6. We can learn two conclusions from Table 6:
>
> (1) LiNGAM suffers a significant decrease with increasing nonlinear ratio because it can only work on linear & non-Gaussian.
>
> (2) The empirical results of CaPS is consistently better than LiNGAM under non-Gaussian settings, which show CaPS can effectively support non-Gaussian noise in addtion to theoretically proved Gaussian noise.
>
> **Table 6. CaPS vs LiNGAM under SynER1 with Gumbel noise.**
> | Linear rate | DirectLiNGAM |           |             | CaPS    |         |              |
> |-------------|--------------|-----------|-------------|---------|---------|--------------|
> |             | SHD          | SID       | F1          | SHD     | SID     | F1           |
> | 0           | 8.2±1.7      | 25.8±11.2 | 0.125±0.174 | 0.6±0.8 | 0.6±0.8 | 0.966±0.044  |
> | 0.25        | 6.8±1.7      | 20.8±11.9 | 0.358±0.159 | 1.2±1.6 | 1.8±2.4 | 0.925±0.1    |
> | 0.5         | 6.0±1.4      | 18.6±11.1 | 0.467±0.109 | 2.4±1.9 | 3.8±3.8 | 0.842±0.130  |
> | 0.75        | 3.6±2.1      | 10.4±8.8  | 0.735±0.174 | 1.2±1.1 | 2.0±1.8 | 0.927±0.075  |
> | 1           | 2.4±1.5      | 7.6±9.4   | 0.844±0.097 | 0.8±0.7 | 2.0±2.6 | 0.944±0.050  |

---

### Official Review · Reviewer_wbDM · 2024-07-22

**Soundness:** 3
**Presentation:** 2
**Contribution:** 1
**Rating:** 4
**Confidence:** 4

**Summary:**

This paper addresses the challenge of ordering-based causal discovery, which involves first determining the topological ordering of variables (typically by recursively identifying sub-leaf nodes) and then identifying the parent set for each variable.

Existing methods often focus on either nonlinear or linear relationships. For instance, SCORE relies on a constant score Jacobian, which fails in the absence of nonlinear relationships, whereas LISTEN employs a precision matrix, which makes no sense in nonlinear contexts.

This work proposes an ordering-based method that accommodates both linear and nonlinear relationships. Specifically, it identifies the topological ordering using the expectation (instead of the variance) of the score's Jacobian, under a sortability assumption on the exogenous noise components. Subsequently, average treatment effect estimation is extended to identify the parent sets.

**Strengths:**

1. The application of ordering-based causal discovery methods to models with both linear and nonlinear relationships is novel to me.

2. The theorems and mathematical details appear to be correct, though I haven't checked all the details.

3. The experimental results are comprehensive, covering various competitors, different settings, and cases where assumptions are violated (e.g., C.7).

**Weaknesses:**

1. **Assumptions are too strong:**
  - For linear relationships in the ANM, additional assumptions are required for identifiability. This paper adopts assumptions similar to those in LISTEN, namely, that the variances of exogenous additive noise components follow the same topological ordering of the causal DAG, akin to VAR-sortability assumptions (Reisach et al., 2021). These assumptions are overly stringent, impractical, and lack testability. More discussions regarding this can be referred to "Structure Learning with Continuous Optimization: A Sober Look and Beyond".
  - The authors also assume that all additive noise components are zero-mean Gaussian. It is unclear if this assumption is utilized throughout the paper or why it was mentioned if not. Also, are these assumptions testable?
  - Regarding the Gaussian assumption, if it is not used for any proof, the authors might consider assuming non-Gaussian noise. This would allow the DAG to be identifiable even with linear relationships. Then, with score matching (which still works, as in Sec4.3 in the SCORE paper) and some straightforward processing (to preserve for non-Gaussianity/residual independence), the problem might still be solvable in a much more elegant way.

2. **Insufficient motivation for "parent score":** Once the topological ordering of the DAG is identified, one could use conditional independence tests between variables and all preceding variables to determine each edge's existence (as in most permutation-based methods), or employ sparse regression, as suggested in the original CAM paper. The authors need to justify the necessity of proposing a "parent score," which seems over-complicated with average treatment effect estimation framework. Are there any advantages (e.g., in terms of time complexity or finite sample guarantee, as in the LISTEN paper)?

3. **Lack of technical novelty:** The technical contributions mainly combine ideas from SCORE and LISTEN, making the results and derivations (e.g., from constant variances to expected value of variances) straightforward extensions of previous work. While novelty is not a primary concern for me, it is worth noting as a minor weakness.

**Questions:**

My major concerns and questions are listed above in "Weaknesses" section.

---

> ### Author Rebuttal · Authors · 2024-08-06
>
> **Preliminaries**
>
> Thank you very much for your valuable comments. Before answering the three question about assumptions, we want to highlight some facts of ANM. Without interventional data, all ANM-based models ($y=f(x)+\epsilon$) have to make some assumptions on $f$ or $\epsilon$ due to the problem of backward model (see Prop.23 in ref. [9]). Some paper make purely linear or nonlinear assumption on $f$, e.g., SCORE and LISTEN. This paper does not make any additional assumptions on $f$ (linear or nonlinear), therefore, some assumptions on $\epsilon$ are inevitable. To the best of our knowledge, our assumptions is the weakest assumption that works well under both linear and nonlinear ANM.
>
> **Q1.1:** VAR-sortability assumption are too strong.
>
> **A1.1:** We need to emphasise that CaPS works under (i) **or** (ii) in assumption 1. So, our assumption is **weaker** than VAR-sortability. CaPS also can work well using the condition (ii) if VAR-sortability does not hold. For example, considering a VAR-unsortable scenario with $\sigma^2\sim U(1,10)$ and causal effect greater than 0.9, CaPS can also work well because the the sum of parent score is greater than the given lower bound in (ii). Experimentally, the results in the App. C.7 also support the conclusion that CaPS works well even without VAR-sortability.
>
> **Q1.2:** Is zero-mean Gaussian utilized? testable?
>
> **A1.2:** This zero-mean Gaussian is a de facto standard in all ANM-based method (see ref. [9,11,13,...]), as this setting does not lose any generalization. Any ANMs with non-zero-mean Gaussian $\epsilon_n \sim N(\mu,\sigma)$ is equivalent to a ANM with zero-mean Gaussian $\epsilon_z \sim N(0,\sigma)$, because $f(x)+\epsilon_n$$=f(x)+\mu+\epsilon_z$$=F(x)+\epsilon_z$. The non-zero-mean Gaussian can be considered as $f$ with different bias in ANM. As we already provided the performance under zero-mean Gaussian noise in Fig. 2, we additionally test the performance under non-zero-mean Gaussian in Table 1 and 2 to further address your concerns. The conclusion is that $\mu$ does not significantly affect the relative performance of all compared baseline and CaPS still achieves the best performance.
>
> **Q1.3:** non-Gaussian proof? like Sec4.3 in the SCORE.
>
> **A1.3:** As mentioned in the preliminaries of this rebuttal, SCORE can extend to non-Gaussian because they make strong assumption on $f$ (purely nonlinear) and highly rely on the nonlinear properties for the proof. This paper handles a scenario with more generalised relations without any addtional linear or nonlinear assumption, so we need to use some properties of $\epsilon$. Theoretically, the Gaussian assumption is necessary, which are used in Eq. 10 to prove Theorem 1. However, experimentally, App. C.7 shows that CaPS performs consistently well under non-Gaussian noise (Gumbel and Laplace), which shows the potential of CaPS generalization to non-Gaussian noises.
>
> Thanks for your inspiring suggestion. We are still working to find a more elegant solution to relax the $\epsilon$ to non-Gaussian noise. However, we believe that CaPS, as the first ordering-based method capable of handling both linear and nonlinear, contributes sufficiently to the field.
>
> **Q1.4:** motivation for parent score.
>
> **A1.4:** Before we recall the motivation of parent score, we want to remind that parent score can be directly decoupled from the score’s Jacobian using Algorithm 1, which does not introduce additional computational complexity. The average treatment effect framework you mentioned is only used for theoretical interpretation. Here is the advantages of introducing parent score.
>
> (1) More clear theoretical interpretation. Why condition (ii) is a lower bound of identifiable causal effects? What's the physical meaning of Theorem 1? It is hard to to explain without parent score and the given average treatment effect framework. However, we can clearly answer this questions with parent score in the Corollary 1 (see details in lines 189-201 and App. A.5), i.e., the left-hand side of condition (ii) is sum of causal effect and Theorem 1 is finding a leaf node with weakest causal effect.
>
> (2) Better performance and speed. Parent score can help to accelerate the pruning process and correct inaccurate predictions in the pruning step. With this post-processing, the performance can be further improve in real-world dataset (12.6% of Sachs and 3.6% for Syntren in F1). More importantly, it can largely accelerate the pruning process especially when nodes are increasing (78.8% for 50 nodes and 49.41 for 20 nodes; see Fig.8). This make CaPS have better capablity of handling large-scale structures.
>
> Therefore, parent score is not a over-complicated design.
>
> **Q1.5:** lack of novelty. CaPS combine ideas from SCORE and LISTEN?
>
> **A1.5:** We understand your concerns, but CaPS is totally different from SCORE and LISTEN the following reasons:
>
> (1) More generalized scenarios. CaPS aims to handle both linear and nonlinear and most possibly mixed relations while SCORE only handles nonlinear relations and LISTEN only handles linear relations. CaPS is the first ordering-based method to deal with this scenario.
>
> (2) Different theoretical ideas. As mentioned in the preliminaries of the rebuttal, SCORE and LISTEN is derived through the properties of nonlinear $f$ while CaPS derives Theorem 1 by the properties of $\epsilon$, which is a totally different ideas in theoretical perspective (see details in App. A.2). Therefore, although the theorem only different on "expectation" and "variance", the derivations is totally different and can not be straightforwardly extended whether from SCORE or LISTEN.
>
> (3) New concept for identifiability and post-processing. This paper proposes a new concept of "parent score", which gives a new lower bound for identifiability with causal effect (see Corollary 1). This concept can be used to accelerate the pruning process (see Fig. 8) and correct inaccurate predictions in the pruning step (see Fig. 4 and 7).

---

> ### Author Response · Authors · 2024-08-06
> **Table 1&2: experiment of non-zero-mean Gaussian noise**
>
> **Table 1. SynER1 with $\epsilon \sim N(1,\sigma^2)$**
> | Dataset     | SynER1 ($\mu$=1) |             |             |             |             |              |
> |-------------|---------------|-------------|-------------|-------------|-------------|--------------|
> | Linear rate | Metrics       | COLEM       | CAM         | SCORE       | DiffAN      | CaPS         |
> | 0           | SHD           | 6.4±0.8     | **0.4±0.4**     | 0.6±0.8     | 2.4±1.2     | 0.6±0.8      |
> |             | SID           | 19.4±8.2    | **1.6±2.7**     | 2.6±4.7     | 10.2±5.3    | 4.2±7.9      |
> |             | F1            | 0.403±0.134 | 0.967±0.044 | **0.969±0.040** | 0.788±0.073 | 0.958±0.061  |
> | 0.25        | SHD           | 5.6±1.8     | 1.0±1.5     | 1.0±0.63    | 2.6±1.0     | **0.8±0.7**      |
> |             | SID           | 19.0±10.4   | **2.4±3.4**     | 5.0±5.7     | 13.6±8.3    | 5.6±7.6      |
> |             | F1            | 0.489±0.215 | 0.911±0.119 | 0.919±0.092 | 0.760±0.114 | **0.945±0.05**7  |
> | 0.5         | SHD           | 4.6±3.1     | 2.2±1.3     | 2.4±1.9     | 2.6±1.6     | **0.8±0.7**      |
> |             | SID           | 20.2±17.3   | 6.8±3.9     | 9.8±8.7     | 7.0±4.5     | **2.4±2.7**      |
> |             | F1            | 0.582±0.275 | 0.799±0.108 | 0.788±0.155 | 0.788±0.112 | **0.932±0.064**  |
> | 0.75        | SHD           | 4.2±1.73    | 3.4±1.9     | 3.2±2.9     | 3.8±3.7     | **0.8±1.1**      |
> |             | SID           | 22.0±10.1   | 16.8±12.1   | 7.6±8.2     | 8.2±7.7     | **3.0±4.2**      |
> |             | F1            | 0.612±0.291 | 0.663±0.198 | 0.739±0.252 | 0.740±0.213 | **0.924±0.098**  |
> | 1           | SHD           | 3.4±3.0     | 4.0±2.0     | 4.2±3.4     | 5.0±3.1     | **1.0±1.0**      |
> |             | SID           | 10.6±10.5   | 18.0±11.8   | 12.0±10.6   | 12.0±6.2    | **3.2±4.1**      |
> |             | F1            | 0.746±0.201 | 0.625±0.199 | 0.655±0.286 | 0.631±0.180 | **0.913±0.091**  |
>
> **Table 2. SynER1 with $\epsilon \sim N(10,\sigma^2)$**
> | Dataset | SynER1 ($\mu$=10) |             |             |             |             |              |
> |---------|----------------|-------------|-------------|-------------|-------------|--------------|
> | Linear rate | Metrics       | COLEM       | CAM         | SCORE       | DiffAN      | CaPS         |
> | 0       | SHD            | 6.6±0.8     | **0.4±0.4**     | **0.4±0.4**     | 3.2±0.7     | 0.6±0.8      |
> |         | SID            | 19.6±8.4    | **1.6±2.7**     | 1.8±3.1     | 12.8±3.8    | 4.2±7.9      |
> |         | F1             | 0.386±0.118 | 0.967±0.044 | **0.978±0.025** | 0.684±0.063 | 0.958±0.061  |
> | 0.25    | SHD            | 5.8±1.9     | 1.0±1.5     | 1.0±0.6     | 4.0±1.6     | **0.8±0.7**      |
> |         | SID            | 19.2±10.5   | **2.4±3.4**     | 5.2±5.1     | 15.6±10.2   | 5.6±7.6      |
> |         | F1             | 0.472±0.212 | 0.911±0.119 | 0.900±0.088 | 0.623±0.166 | **0.945±0.057**  |
> | 0.5     | SHD            | 4.6±3.0     | 2.2±1.3     | 2.2±1.6     | 4.6±1.0     | **0.6±0.8**      |
> |         | SID            | 20.2±17.3   | 6.8±3.9     | 8.8±6.4     | 10.0±2.8    | **1.6±2.7**      |
> |         | F1             | 0.582±0.27  | 0.799±0.108 | 0.808±0.119 | 0.654±0.046 | **0.961±0.055**  |
> | 0.75    | SHD            | 4.3±0.80    | 3.4±1.9     | 2.0±1.7     | 3.4±2.2     | **0.8±1.1**      |
> |         | SID            | 25.0±18.4   | 16.8±12.1   | 7.4±7.5     | 10.4±5.8    | **3.0±4.2**      |
> |         | F1             | 0.611±0.162 | 0.663±0.198 | 0.808±0.184 | 0.751±0.110 | **0.924±0.098**  |
> | 1       | SHD            | 3.0±3.7     | 4.0±2.0     | 3.2±2.9     | 5.4±2.4     | **1.2±1.1**      |
> |         | SID            | 11.2±8.2    | 18.0±11.8   | 11.4±10.4   | 13.0±5.9    | **3.8±4.0**      |
> |         | F1             | 0.769±0.290 | 0.625±0.199 | 0.716±0.248 | 0.613±0.130 | **0.892±0.093**  |

---

> ### Comment · Reviewer_wbDM · 2024-08-12
>
> Thank the authors for the detailed response.
>
> For the assumptions, technically yes, they are slightly weaker than VAR-sortability with another condition allowed. However, the another condition, just similar to VAR-sortability, is mainly human-crafted for the framework. It lacks any natural theoretical interpretability or practical testability.
>
> For zero-mean Gaussian, thanks for the additional experiments, but "$\mu$ does not significantly affect the relative performance" -- could the authors please confirm that whether zero-mean is needed for the asymptotic identifiability guarantee?
>
> For Gaussian noise assumption, "it is used in Eq. 10 to prove Theorem 1" -- however, I couldn't see from the proof on where specifically a Gaussian distribution is needed. Instead, I can only see the use on general forms on means and variances, together with the assumptions. But in any way, if only Gaussian noise is allowed, it would be a further shortcoming for this work: the identifiability will be unclear, and the the method will be less practical.
>
> For motivation of parental score, thanks for reminding me that it does not introduce additional computational complexity. Though not theoretical interesting, it indeed offers empirical gain. I have adjusted my score to reflect this point.

---

> > ### Author Response · Authors · 2024-08-13
> > **Thanks for your feedback**
> >
> > Thank you for your insightful comments and for adjusting the scores. We would like to clarify a few points where we believe there might have been a misinterpretation of our work.
> >
> > For condition (ii), we already provide its theoretical interpretability in Corollary1 and App. A.5. This condition is a straightforward lower bound for identifiable causal effect, which demonstrates **for the first time** how strong causal effects can be recognized. For practical testability of condition (ii), as given in Q&A 2.3 of Reviewer 8ao9, **most of conditions is unverifiable without the ground truth SEM (even simple conditions like linear and nonlinear)**. Therefore, to further provide more practical testability, additional experiments are given on synthetic data. We test different setting of the sum of causal effect to show the performance of CaPS when condition (ii) are perfectly satisfied / likely satisfied / likely unsatisfied. In order to accurately control the causal effect and the lower bound in condition (ii), we use the linear SynER1 with the noise standard deviation in $U(0.4, 0.8)$. Under this settings, condition (ii) will perfectly satisfied when the minimal causal effect is geater than $\sqrt{0.8^2(\frac{1}{0.4^2}-\frac{1}{0.8^2}))}=\sqrt{3}$ because the node with weakest SATE and single child will greater than the theoretical lower bround. The experimental results are shown in Table 5, which gives the practical testability and shows that **CaPS will work well when condition (ii) are perfectly satisfied and likely satisfied**.
> >
> > **Table 5. practical testability of condition (ii) on SynER1**
> > | condition (ii)      | causal effect | SHD     | SID       | F1           |
> > |---------------------|---------------|---------|-----------|--------------|
> > | perfectly satisfied | $U(1.8, 2.0)$   | 0.0±0.0 | 0.0±0.0   | 1.000±0.000  |
> > | likely satisfied    | $U(1.6, 1.8)$   | 0.0±0.0 | 0.0±0.0   | 1.000±0.000  |
> > | likely satisfied    | $U(1.4, 1.6)$   | 0.2±0.4 | 0.8±1.6   | 0.975±0.050  |
> > | likely satisfied    | $U(1.2, 1.4)$  | 0.4±0.4 | 1.4±1.7   | 0.964±0.049  |
> > | likely unsatisfied  | $U(0.6, 0.8)$   | 2.6±1.0 | 10.0±5.6  | 0.772±0.065  |
> > | likely unsatisfied  | $U(0.4, 0.6)$   | 4.2±1.4 | 8.8±6.8   | 0.689±0.082  |
> > | likely unsatisfied  | $U(0.2, 0.4)$   | 4.0±1.0 | 15.8±11.7 | 0.655±0.103  |
> >
> > For zero-mean Gaussian, this settings is widely used in previous work (ref. [9,11,13,19,...]) because the $\epsilon$ are usually consider as the residual of $f(pa(x))$. As LISTEN have pointed out, "without loss of generality, we assume that $E(X_i)=E(N_i)=0$". This is because the non-zero-mean and zero-mean is equivalent in a ANM, which we already explained in Q&A 1.2. Thus, any non-zero-mean ANM with $\epsilon_n \sim N(\mu,\sigma)$ can transform to a zero-mean ANM with $\epsilon_z \sim N(0,\sigma)$ then follow the same derivation. And that's why $\mu$ does not significantly affect the relative performance empirically in Table 1 & 2.
> >
> > For Gaussian noise assumption, to be precise, we use Gaussian pdf for derive Eq.2 and Eq.10 comes from Eq.2. Since CaPS **handles linear, nonlinear and even mixed relations simultaneously**, we put almost no restrictions on $f$ in ANM. Under this premise, as CaPS handles  both types of relations for the first time, it seems too strict to require a solution for both linear & nonlinear and Gaussian & non-Gaussian at the same time. Although this is the weakest condition we can derive for both linear and nonlinear scenario, the experimental results in App. C.7 have been encouraging in cases of non-Gaussian noise. We are striving to broaden it to more relaxed conditions.
> >
> > Once again, thanks for your careful review and the time you have dedicated to our manuscript. We hope that our responses will address your concerns.

---

### Decision · Program_Chairs · 2024-09-25

**Decision:**

Accept (poster)

**Comment:**

Reviewers agreed that the paper introduces a novel method, extending the capabilities of causal discovery. There were concerns about the assumptions necessary for the method to work, but as the authors point out, all causal discovery methods must by necessity make some difficult assumptions, and the proposed method weakens some important assumptions made in previous work. Reviewers commended the extent of the experimental validation, with the method overall achieving encouraging results.